# The global diabatic circulation of the stratosphere as a metric for the Brewer-Dobson Circulation

Marianna Linz[1], Marta Abalos[2], Anne Sasha Glanville[3], Douglas E. Kinnison[3], Alison Ming[4], and Jessica L. Neu[5]

[1]Department of Atmospheric and Oceanic Sciences, University of California, Los Angeles, 520 Portola Plaza Los Angeles, CA 90095, USA
[2]Department of Earth Physics and Astrophysics, Universidad Complutense de Madrid, Madrid, Spain.
[3]Atmospheric Chemistry Observations and Modeling Laboratory, National Center for Atmospheric Research, Boulder, CO, USA
[4]British Antarctic Survey, Cambridge, CB3 0ET, UK
[5]Jet Propulsion Laboratory, California Institute of Technology, 4800 Oak Grove Dr, Pasadena, CA, 91109 USA

*Correspondence to:* Marianna Linz (mlinz@atmos.ucla.edu)

**Abstract.** The circulation of the stratosphere, also known as the Brewer-Dobson circulation, transports water vapor and ozone, with implications for radiative forcing and climate. This circulation is typically quantified from model output by calculating the tropical upwelling vertical velocity in the residual circulation framework, and it is estimated from observations by using time series of tropical water vapor to infer a vertical velocity. Recent theory has introduced a method to calculate strength of the global mean diabatic circulation through isentropes from satellite measurements of long-lived tracers. In this paper, we explore this global diabatic circulation as it relates to the residual circulation vertical velocity, stratospheric water vapor, and ozone at interannual timescales. We use a comprehensive climate model, three reanalysis data products, and satellite ozone data. The different metrics for the circulation have different properties, especially with regards to the vertical autocorrelation. In the model, the different residual circulation metrics agree closely and are well correlated with the global diabatic circulation, except in the lowermost stratosphere. In the reanalysis products however, there are more differences throughout, indicating the dynamical inconsistencies of these products. The vertical velocity derived from the time series of water vapor in the tropics is significantly correlated with the global diabatic circulation, but this relationship is not as strong as that between the global diabatic circulation and the residual circulation vertical velocity. We find that the global diabatic circulation in the lower to middle stratosphere (up to 500 K) is correlated with the total column ozone in the high latitudes and in the tropics. The upper level circulation is also correlated with the total column ozone, primarily in the subtropics, and we show that this is due to the correlation of both the circulation and the ozone with upper level temperatures.

*Copyright statement.* TEXT

# 1 Introduction

The Brewer–Dobson circulation (BDC) is important for the distribution of trace gases in the stratosphere (Butchart, 2014) including water vapor, the radiative effects of which have been shown to impact surface climate (Dessler et al., 2013), and ozone, which impacts tropospheric circulation (e.g., Polvani et al., 2011) and human health (e.g. Abarca and Casiccia 2002).

In models and reanalysis products, the BDC is frequently quantified by the vertical velocity in the Transformed Eulerian Mean (TEM) framework (Andrews et al., 1987), averaged over the well-mixed tropics (e.g. Butchart et al., 2006; Li et al., 2008; Seviour et al., 2012; Hardiman et al., 2017). In steady state, the total upwelling and downwelling mass fluxes must be equal, and so characterizing the tropics alone is considered sufficient. The TEM framework provides formalism that approximates the Lagrangian-mean mass transport, and in the limit of adiabatic, small-amplitude eddies, the TEM residual mean circulation

is equivalent to the density-weighted isentropic mean circulation. Multimodel comparisons (Butchart et al., 2010) and inter-reanalysis comparisons (Abalos et al., 2015; Kobayashi and Iwasaki, 2016) have used the residual mean circulation at 70 hPa, averaged in the tropics, as a metric to evaluate the mean and trends of the BDC. The 70 hPa level is consistently within the stratosphere even in climate models that do not accurately simulate tropopause height. As it is in the lower stratosphere, it approximates the mass flux between the troposphere and stratosphere and, as such, is related to the water vapor flux and ozone

transport.

Models predict that the residual mean circulation through a given pressure surface will increase in the future by about 2% per decade in the lower stratosphere and about 1% per decade in the middle and upper stratosphere (Butchart et al., 2010). This is a natural consequence of the lifting of the atmospheric circulation (e.g. Singh and O'Gorman, 2012; Oberländer-Hayn et al., 2016), and there are also dynamical reasons why one might expect a true acceleration of the BDC (e.g. McLandress and

Shepherd, 2009; Shepherd and McLandress, 2011; Garny et al., 2011). However, observations have not shown such a robust trend (e.g. Engel et al., 2017; Stiller et al., 2012; Haenel et al., 2015). This disagreement can be attributed partially to the large internal variability in the system that prevents a 2% per decade trend from being detected without 30 years of data (Hardiman et al., 2017), and partially to the fact that there is no truly "like-to-like" comparison; a modeled tracer that is sampled like the observations can also fail to show a trend even when such a trend exists in the model (Garcia and Randel, 2008). Models

also show that polar ozone loss has dampened the acceleration of the circulation, with an asymmetric effect on the different hemispheres (M. Polvani et al., 2017).

The TEM vertical velocity, which shows a robust trend in models, is a useful metric for understanding stratospheric dynamics. However, apart from its theoretical relationship with the Lagrangian-mean mass transport, it is not straightforward to relate the TEM vertical velocity to the tracer transport that is so important to climate due to the presence of other transport processes

such as mixing (Dietmüller et al., 2017, 2018; Ray et al., 2010, 2016). In contrast, the global average diabatic overturning circulation through isentropes can be theoretically related to observed tracer distributions through the idealized tracer "age of air" (Neu and Plumb, 1999; Linz et al., 2016). This global diabatic circulation has been calculated from two different satellite data products (Linz et al., 2017), thus motivating the use of the global diabatic circulation as a metric for the BDC strength in addition to the TEM vertical velocity. In this paper, we explore differences between the global diabatic circulation and other

calculations for the strength of the circulation in order to understand the relationship of this new constraint to more common metrics.

The calculation of the global diabatic circulation in Linz et al. (2017) is the first of its kind, but not the first observational estimate of the stratospheric circulation strength. Water vapor is transported into the stratosphere through the cold tropical tropopause, which has a strong seasonal cycle in temperature. The resulting time series of water vapor at the cold-point tropopause similarly has a strong seasonal cycle. By tracking the upward movement of the dry and wet phases over time, the water vapor signal—which is nearly conserved above the cold point tropopause—can be used to calculate an effective velocity ($w_{TR}$). "Effective" refers to the aggregated transport, which includes the effects of advection and mixing. As a result, this "water vapor tape recorder" (Mote et al., 1996) method must be used with caution when studying the tropical tropopause layer (Podglajen et al., 2017) and with even more caution when comparing models (Dietmüller et al., 2018). This study minimizes such issues by focusing on the region above the tropical tropopause layer and by using a zonal mean between 10°S-10°N to reduce the influence of horizontal mixing between the subtropics and the midlatitudes. We will explore the relationship between the vertical velocity derived from water vapor in the deep tropics and the global diabatic circulation.

One of the primary motivations for studying the BDC and its variability is its influence on stratospheric ozone. The circulation is known to transport ozone—this is why Dobson proposed it in the first place (Dobson et al., 1929), even if he concluded that this circulation was far-fetched. While the qualitative description of the influence of the stratospheric circulation on ozone variability is well established—variations in the transport of ozone from its maximum production location in the middle stratosphere in the tropics to the midlatitudes and poles—quantifying this effect is not simple. Furthermore, the interplay between the temperature, ozone and circulation can lead to complex feedbacks. We know from observational studies that changes to dynamical quantities impact polar ozone (Hassler et al., 2011), and that the ozone hole recovery is currently being modulated by the dynamics (Solomon et al., 2016). In turn, variability and trends in the ozone affect the circulation (e.g. Polvani et al., 2011; Bandoro et al., 2014). In the Northern hemisphere, the variability in hemispherically averaged upward Eliassen-Palm (EP) flux at 100 hPa from the early NCEP reanalysis data product has been shown to explain about 50% of the interannual variability of total column ozone in wintertime (Fusco and Salby, 1999) with the influence of the wave driving dependent on the latitude (Reinsel et al., 2005). These strong relationships are a motivating factor in using the TEM residual mean vertical velocity, which is directly related to the EP flux, as a metric for the BDC strength. The global diabatic overturning circulation on isentropes has been demonstrated to be related to tracer transport more directly, but its relationship with ozone is unknown.

This paper serves to explore the global diabatic circulation as a metric for the stratospheric circulation strength. Section 2 describes the model runs, reanalysis products, satellite data, and regression methods. In Section 3, we provide an explanation of the steps for calculating the global diabatic circulation on isentropes, the necessary model output, and its advantages and disadvantages. In Section 4, we examine three different calculations for the TEM vertical velocity, including the underlying assumptions, and with different tropical averaging. Then we compare the global diabatic circulation to the more traditionally used TEM vertical velocity calculated in these three different ways (Abalos et al., 2015) for three different reanalysis products and for the Whole Atmosphere Community Climate Model (WACCM). Thus we determine how the information provided by this new metric compares to the information more typically used. We find close agreement between the global diabatic

circulation strength and one of the three calculation methods for the TEM vertical velocity for the reanalyses (regardless of averaging choice), and close agreement between the global diabatic circulation strength and all three calculations for the TEM vertical velocity in the model (though only with fixed-latitude tropics). In Section 5, we compare the tropical vertical velocity calculated from the water vapor tape recorder (Niwano et al., 2003) from the WACCM model to the total overturning circulation. Similar to the good agreement found for the modeled residual circulation and global diabatic circulation, the global diabatic circulation strength and the water vapor tape recorder are closely linked in the model, although the correlation is weaker. In Section 6, we examine the relationship between the diabatic overturning circulation and stratospheric ozone, using data, reanalyses, and WACCM. We find that the lower branch of the circulation is important for polar ozone, while the upper branch is the most important for subtropical ozone. The latter relationship is driven by the temperature dependence of the photochemistry and covariance of temperature and the global diabatic circulation. The ozone results are consistent with known relationships between TEM vertical velocity and ozone, demonstrating that the global diabatic circulation is as good a metric for ozone variability. Section 7 summarizes the results and discusses implications and future work.

## 2   Model, reanalysis products, satellite data, and methods

A summary of the products, their resolutions, and associated references is given in Table 1.

For the model in this study, we use the Whole Atmosphere Community Climate Model (WACCM), a state of the art, chemistry-climate model. This model uses the physical parameterizations and finite-volume dynamical core (Lin, 2004) from the Community Atmosphere Model, version 4 (Neale et al., 2013), with a vertical extent from the surface to the lower thermosphere, and 31 pressure levels from 193 hPa to 0.3 hPa. The WACCM simulation is the first member of an ensemble run based on the Chemistry Climate Model Initiative REF-C1 scenario (Morgenstern et al., 2017), and is forced with prescribed observed sea surface temperatures. This model simulation was shown to have a global diabatic circulation strength that agrees closely with the satellite tracer observations at 460 K (Linz et al., 2017). This study covers the time period from 1980–2014.

Three different renalysis data products are used in this study, following upon the work by Abalos et al. (2015) and Linz et al. (2017). These are the ECMWF Reanalysis Interim (ERA-Interim, Dee et al. 2011), the Modern Era Retrospective analysis for Research and Applications (MERRA, Rienecker et al. 2011), and the Japanese 55-year Reanalysis (JRA55, Kobayashi et al. 2015). Reanalyses are used for the same time period as WACCM, for consistency in the comparisons. These reanalyses are based on assimilation of different sets of data into three different models and using different assimilation schemes, leading to some significant differences in their output, especially above 10 hPa. Beneath 10 hPa, Abalos et al. (2015) showed that more uncertainty arose from the choice of method of calculation of the vertical velocity than from the choice of reanalysis, concluding that differences between reanalyses were relatively small (except for trends). We build upon that result here and suggest that because of uncertainties in radiative heating rates, the reanalyses are not as robust in certain contexts.

Finally, we consider the total column ozone measurements from the Solar Backscatter Ultraviolet Instrument (SBUV) from 1980–2013 from the version 8.6 SBUV ozone data record (McPeters et al., 2013). This data is based on nine recalibrated

| Data source | Resolution | Reference |
|---|---|---|
| WACCM | 2.5 ° lon, 1.875 ° lat, 31 pressure levels from 193 hPa to 0.3 hPa | Marsh et al., 2013, Garcia et al. 2017 |
| ERA-Interim | 1°×1°, 26 pressure levels from 150 hPa to 0.5 hPa | Dee et al. 2011 |
| JRA55 | 1.25°×1.25°, 16 pressure levels from 225 hPa to 1 hPa | Kobayashi et al. 2015 |
| MERRA | 1.25°×1.25°, 17 pressure levels from 200 hPa to 0.5 hPa | Rienecker et al. 2011 |
| SBUV $O_3$ | zonal mean, 5 ° lat, total column | McPeters et al. 2013 |

**Table 1.** Model output, reanalysis products, and ozone data used in this study.

SBUV instruments with total column ozone measurements that are consistent with ground-based observations of total column ozone to within 1%. We use the total column ozone as it has the least uncertainty for use in long term correlation calculations.

As the primary motivation of this paper is to evaluate relationships between the dynamical and tracer quantities, it utilizes correlations extensively. The standard Pearson correlation coefficient is reported for each relationship. Only results signif-
5 icant at 95% level or greater are reported. Time series are deseasonalized by removing the climatology of each variable. Cross-correlations are used to examine the differences in the vertical structures of the different quantities. When these cross-correlations are between transport metrics that have different vertical coordinates, a climatological relationship between tropical (20°S-20°N) potential temperature and pressure is shown.

In the stratosphere, correlations of circulation metrics might be expected to be driven by the Quasi-Biennial Oscillation
(QBO) in addition to the seasonal cycle. Rather than explicitly removing this, we account for it by examining filtered time series and coherence (e.g. Figure 1) and highlight the cases where this is important. Many of the relationships examined are coherent at timescales shorter than the 2-3 year QBO period, though coherence is particularly high at that period. The relationship of dynamical variables with trace gases have less high frequency variability, and therefore tend to be dominated by the QBO.

## 3 Calculating the global diabatic circulation on isentropes

Why would we need a different metric for the BDC? The residual mean tropical upwelling at 70 hPa has been used for at least a decade (Butchart et al., 2006). However, it is neither directly observable nor easily relatable to observations. A metric for models and reanalyses ideally should be able to be constrained by data. The Tropical Leaky Pipe Model (Neu and Plumb, 1999), a three-box model of the stratospheric circulation that treats the tropics as largely isolated from the extratropics, results
in the conclusion that the difference between midlatitude age and tropical age is related to the circulation. Linz et al. (2016) translated this model into isentropic coordinates to show a direct relationship between the idealized age of air (Hall and Plumb, 1994) and the diabatic circulation through an isentropic surface, demonstrating that the difference between the age of air that is downwelling and the age of air that is upwelling through each isentrope is inversely proportional to the diabatic mass flux through that surface, in statistically steady state and neglecting diabatic diffusion. Thus, the global diabatic circulation through

an isentrope reflects the total tracer flux and should be considered an alternative, or at least additional, metric. This global diabatic circulation can also be calculated from satellite data.

## 3.1 Definition of the global diabatic circulation

The time-dependent, global, diabatic overturning mass flux through an isentrope is defined to be the average of the upwelling
and downwelling mass fluxes, as follows.

As in Linz et al. (2016), we define the total upwelling mass flux, $\mathcal{M}_u$, and the total downwelling mass flux $\mathcal{M}_d$, through an isentropic surface:

$$\mathcal{M}_u = \int_{up} \sigma \dot{\theta} dA, \text{ and} \tag{1}$$

$$\mathcal{M}_d = - \int_{down} \sigma \dot{\theta} dA. \tag{2}$$

$\dot{\theta}$ is the total diabatic heating rate, and $\sigma = -g^{-1}\partial p/\partial \theta$ is the isentropic density. The limits of integration are the regions of the isentrope through which air is upwelling ($\dot{\theta} >= 0$) and downwelling ($\dot{\theta} < 0$) instantaneously. Since the monthly mean is not in equilibrium, some amount of storage may take place, and these two will not necessarily be equal. We therefore define the total overturning circulation as the average:

$$\mathcal{M}(\theta) = (\mathcal{M}_u - \mathcal{M}_d)/2. \tag{3}$$

This is an arbitrary but sensible definition. Although one could certainly consider the extratropics or tropics alone, the treatment in (3) accounts for simultaneous variability in the extratropics and in the tropics, thus providing an instantaneous global average overturning circulation strength. Furthermore, it is this quantity that is directly related to the age of air distribution (Linz et al., 2016).

A note about the use of isentropic coordinates: the isentropic framework makes separation of the diabatic and adiabatic components completely natural—they are simply the vertical and horizontal motions, respectively. In the annual mean and in steady state, the circulation on isentropes is much the same as the circulation on pressure surfaces. The seasonal variability of circulation on pressure surfaces and on isentropes differs, however. For example, the isentropes descend at the poles during springtime, which leads to upward motion of the air relative to the isentropes. Such springtime polar upwelling is not visible
if pressure surfaces are used instead. Seasonal variability is removed from all time series in this study, and thus we attempt to minimize the effect of this difference on the comparisons. For trends however, the longer-term motion of the isentropes may well be different from the motion of the pressure surfaces, which will be moving up as the tropopause lifts (e.g., Singh and O'Gorman 2012). Thus, we might expect trends to have significantly different results depending on the choice of coordinate system, perhaps as different from trends through pressure surfaces as those calculated relative to the tropopause height
(Oberländer-Hayn et al., 2016).

To calculate the global diabatic circulation from model output or reanalysis, one thus needs the total diabatic heating rate, the temperature, and the pressure. The diabatic heating rate is output differently in different models, but it is straightforward. The diabatic heating rate consists predominantly of two terms, the latent heat flux from phase changes of water and the radiative heating and cooling (Fueglistaler et al., 2009; Wright and Fueglistaler, 2013). For levels wholly within the stratosphere, water vapor concentrations are so low that the former is negligible. Models may output other diabatic terms, such as mixing from parameterized gravity waves; alternatively, they may output a total temperature tendency, which contains all of the necessary information in just one term. Almost all models will output these terms on either pressure or model levels. The diabatic heating rate on those levels must then be interpolated to isentropic levels, for which the temperature and pressure fields are necessary. The isentropic density can be calculated by finite difference in pressure and then interpolated to isentropes as well.

Since eddies serve to predominantly mix adiabatically, they are, naturally, less important for the global diabatic circulation than for the residual circulation. In the conversion from the diabatic vertical velocity on pressure surfaces to the diabatic vertical velocity on isentropes, the covariance of the diabatic vertical velocity and the isentropic levels could nevertheless be important. However, this covariance is small enough that monthly mean temporal resolution is sufficient to accurately calculate the circulation; specifically, in ERA-Interim using monthly means instead of 6-hourly means results in no bias throughout most of the stratosphere and up to a 10% bias at the poles in wintertime, which, as the pole is a small area of the globe, is a much smaller bias on the total overturning mass flux. While many models do output monthly mean eddy fluxes to calculate the residual circulation, others, especially older model runs, do not. Almost all models output shortwave and longwave radiation, and as these are by far the dominant terms in the total diabatic heating rate, this metric can be calculated using models that did not report the necessary terms or have the necessary temporal resolution for the residual circulation vertical velocity calculation. The comparatively minimal data requirements for this metric recommend it for intermodel comparisons.

Although the global diabatic circulation strength is a good indication of the integrated eddy forcing on the circulation, but it does not diagnose which eddies are responsible. Thus, models could get the right circulation from the wrong waves, and indeed, there is reason to expect compensation between resolved and parameterized wave driving (Cohen et al., 2013). Because of this compensation, the analysis of different wave forcing contributions to the BDC in the residual mean framework is also potentially problematic. Finding an appropriate way to relate any BDC metric directly to the tropospheric forcing in a way that could inform model development and tuning is an interesting area of research.

## 3.2 Global diabatic circulation characteristics

The mean value of the global diabatic circulation at 460 K for WACCM is 7.1 $\times 10^9$ kg/s, decreasing to 1.8 $\times 10^9$ kg/s at 1000 K (Linz et al., 2017). The seasonal cycle, which is subsequently removed, is shown in the first panel of Figure 1 for two different levels for the global diabatic circulation from WACCM. The lower stratosphere has a single peak, while the upper stratosphere has a semi-annual cycle as well. This climatology is subtracted to obtain the time series shown in the lower panel of Figure 1. Note that the negative of the anomaly is plotted for the lower level, to enable a clear comparison of these two anticorrelated time series. The different timescales of variability are visible, with an obvious QBO signal and shorter timescale variability. Although the correlation between the upper and lower levels is clear and in phase at QBO timescales, the higher

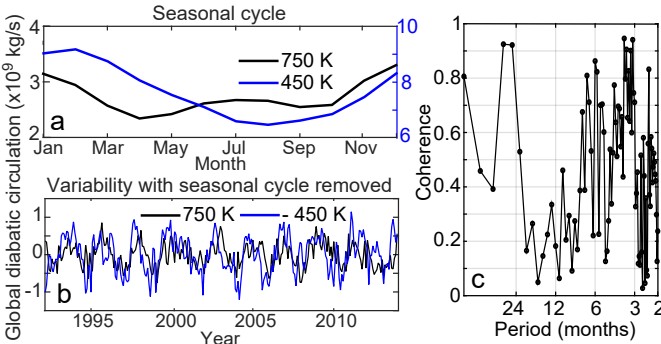

**Figure 1.** The seasonal cycle (a) and interannual variability (b) for the 450 K global diabatic circulation and the 750 K global diabatic circulation in the WACCM model from 1980-2014. Note that in (b) the sign of the anomalies has been reversed for the 450 K level in order to see the agreement. (c) shows the coherence between these two timeseries, demonstrating that the visual correlation evident in (b) is related both to the Quasi-biennial oscillation and to higher frequencies.

frequency variation is also correlated, but with a 20-90 degree phase lag (not shown). The coherence between these two time series is shown in the right panel of Figure 1. There is high coherence at periods of 2-3 years, as expected with the QBO. There is also coherence for periods of shorter than about 9 months, which is unexplained by the QBO.

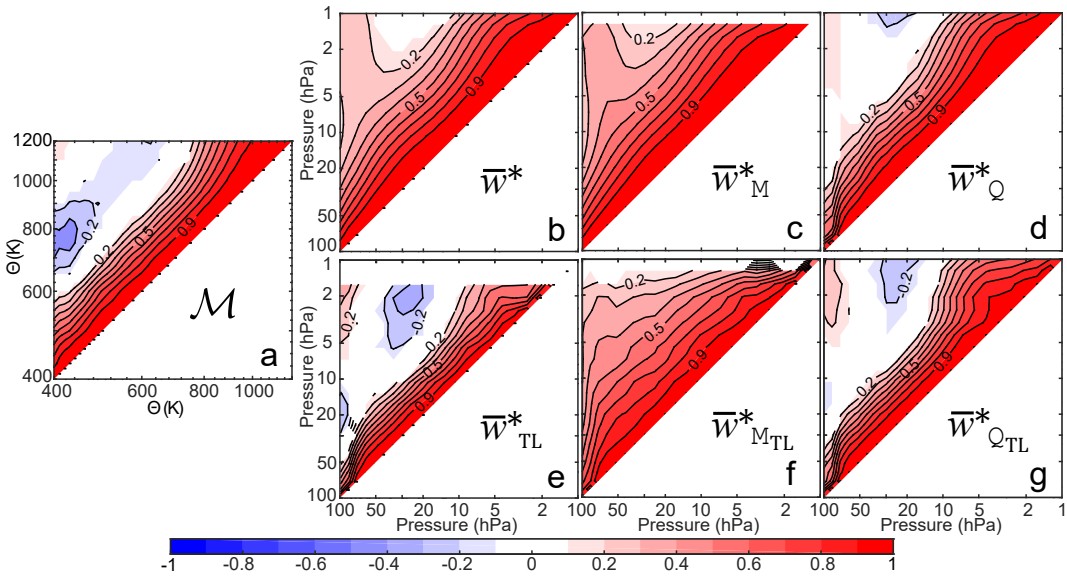

**Figure 2.** Correlation coefficient ($r$) for the autocorrelation of the deseasonalized time series of (a) the global diabatic circulation, and of the three different TEM vertical velocities calculated from WACCM with $30°$ tropics: (b) $\bar{w}^*$, (c) $\bar{w}^*_M$, and (d) $\bar{w}^*_Q$, and with the true turnaround latitudes (e) $\bar{w}^*$, (f) $\bar{w}^*_M$, and (g) $\bar{w}^*_Q$. As the diagonal reflection is redundant, it is not shown. Contours are spaced every 0.1.

The vertical autocorrelation coefficient ($r$) of the global diabatic circulation (with the seasonal variability removed) is shown for WACCM in the first panel of Figure 2. The autocorrelation is relatively narrow in width, so that the variability of the lower level circulation is relatively uncorrelated with that of the upper level circulation. An interesting feature is the weak anticorrelation between lower and upper levels, which can also be seen in the vertical autocorrelation function of the heating rates themselves (in either pressure or isentropic coordinates). Some of this anticorrelation is due to the anticorrelation at the QBO timescales, but the higher frequency variability is also anticorrelated, as can be seen from Figure 1, and the dynamical reasons for this are the subject of ongoing investigation.

### 3.3 Global diabatic circulation trends

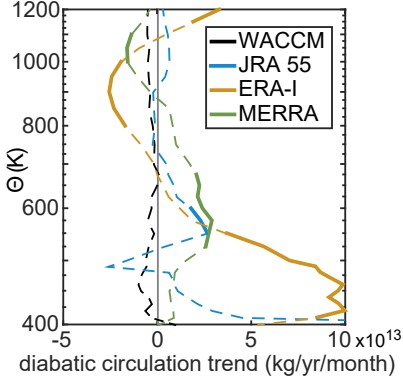

**Figure 3.** Trends in the global diabatic circulation at each level calculated from the three reanalysis data products (JRA55: blue; ERA-Interim: yellow; and MERRA: green) and from the WACCM model (black). Dashed lines show the calculated trends that are not significant at the 95% confidence level, while bolded lines are significant. There are no significant trends in the WACCM model run.

Like the seasonal cycle, the trends in the global diabatic circulation have not previously been examined. We calculate the trends (1980-2014) in the global diabatic circulation from the three different reanalyses and also from the WACCM model run over the same time period, and the results of this are shown in Figure 3. These results are similar to those found by Abalos et al. (2015) for the TEM vertical velocity calculated using the thermodynamic equation, $\bar{w}_Q^*$. Because of the different coordinate system, however, some differences exist. (Note that Abalos et al. 2015 found that the removal of interannual variability does not change the long-term trends significantly.) ERA-Interim shows an acceleration of the lower branch of the circulation and a deceleration of the upper branch. MERRA shows an acceleration around the midstratosphere, where the upper branch begins, and in the uppermost stratosphere. JRA55, meanwhile, only has a small region in the midstratosphere where it shows a statistically significant trend. This is also an acceleration. The WACCM simulation for this time period meanwhile, has no statistically significant trend in the global diabatic circulation at any level, despite significant trends in the thermal structure. This result of no trend in the WACCM overturning is intriguing—although the isentropic levels are changing location over these decades, the total overturning through each isentrope is not significantly changing. This is perhaps related to the lifting

of the circulation described by Oberländer-Hayn et al. (2016), so that the circulation strength is staying the same through isentropes, but moving upwards in pressure. This is consistent with the results of Abalos et al. (2017), who found that trends in the residual streamfunction for a WACCM model run from 1955-2099 were far weaker when calculated with respect to the tropopause (though the trends were still significantly positive over that long period). The differences in trends in this metric compared with the more standard TEM vertical velocity calculation (Andrews et al., 1987), which show significant positive trends at most levels for MERRA and JRA55 regardless of the definition of the tropics (Abalos et al., 2015), demonstrate that although the global diabatic circulation varies closely with the other metrics, trends will appear different, considering that changes to the thermal structure as well as the circulation play a role. Note that since the time series of heating rates in reanalyses are somewhat questionable above 800 K, where they are influenced by changes in the observing systems (Simmons et al., 2014), the trends there are to be treated with caution.

## 4   The global diabatic circulation and TEM vertical mass flux in three reanalyses and a model

The BDC was originally hypothesized to explain observed tracer distributions (Dobson et al., 1929; Brewer, 1949), and therefore the Lagrangian mean transport is, in some sense, the appropriate formalism to study. The TEM residual circulation is not the same as the Lagrangian mean mass transport, as noted explicitly in Andrews and McIntyre (1976). However, under certain conditions (small amplitude, adiabatic eddies), the Lagrangian mean circulation and the TEM residual circulation are identical. The TEM equations also provide unique insight into the forcing of the mean flow by eddies; when the quasigeostrophic approximation holds, the internal forcing of the mean state by the eddies is encompassed by the divergence of the Eliassen-Palm flux (Edmon et al., 1980). Thus, because of the ready interpretation of the wave-mean flow interactions, the TEM residual mean circulation has been the primary diagnostic of the stratosphere for models. It cannot, however, be derived from observations. Here, we try to understand differences in the common methods for calculating the TEM residual circulation vertical velocity and the relationship between it and the global diabatic circulation.

A note about the QBO: although the QBO influences both the residual circulation and the global diabatic circulation, the relationships between the metrics in this section are significantly coherent at all frequencies (see Figure 7 for a comparison of timeseries of $\bar{w}^*$ and $\mathcal{M}$.)

### 4.1   Comparison of TEM vertical velocity calculation methods

Abalos et al. (2015) performed an extensive reanalysis intercomparison of the trends in the TEM vertical mass flux calculated in multiple ways from ERA-Interim, MERRA, and JRA-55. For this work, the calculations were repeated for the WACCM model output. The three different methods for calculating the mass flux are summarized as follows, and for more details see the original paper. The first method is the residual circulation (Andrews et al., 1987), $\bar{w}^*$, in which the residual vertical velocity is calculated based on the Eulerian mean vertical velocity and the meridional eddy heat flux. This method, which we will refer to as the "direct" method relies on performing vertical integrals of the velocity fields from reanalyses. The second calculation of the BDC, $\bar{w}_M^*$, is based on the "downward control" principle (Haynes et al., 1991), and is calculated using momentum balance

equation, integrating the difference of the divergence of the Eliassen-Palm Flux and the zonal mean zonal wind tendencies on surfaces of constant "angular" momentum (in this case, constant latitude) to derive a streamfunction (Randel et al., 2002). The assumption of isolines of angular momentum being equivalent to latitude lines could lead to errors in this estimate. Both of these methods also rely on the applicability of the quasigeostrophic approximation to interpret their results. The final estimate,

$\bar{w}_Q^*$, is calculated by iterating the thermodynamic balance and the continuity equation with no net mass flux across a pressure surface (Murgatroyd and Singleton, 1961; Rosenlof, 1995). Any errors in heating rates will be reflected in this calculation. Because this estimate is also derived from the heating rates, this should be the most closely related to the global diabatic circulation. For this study, we use the deseasonalized timeseries of these estimates of the BDC strength integrated over 30°S–30°N and integrated between the turnaround latitudes (Abalos et al. 2015, Figure 8) at all levels throughout the depth of the

stratosphere.

The first two of these methods both require at least 6-hourly data, while the thermodynamic $\bar{w}_Q^*$ needs only monthly mean data (Lin et al., 2015). For the purposes of this study, the same frequency of data (6-hourly instantaneous values) was used for all three methods and then monthly averages were taken. The interpretations of the results in terms of eddy forcing for both the direct method and the downward control method rely upon quasigeostrophic balance, whereas the thermodynamic

method does not. Thus, we might expect that the two quasigeostrophic, high-frequency data derived estimates would be very similar. Butchart et al. (2006) calculated the mean and the trend in the residual vertical velocity using both methods in a variety of models and found that they were generally similar in magnitude and structure, though differences between the two calculations varied more than differences in the interannual variability of each individual calculation. Rosenlof (1995) compared the thermodynamically calculated streamfunction to the downward-control streamfunction and found them to be

similar, but with the strongest differences in the lower stratosphere. Abalos et al. (2015) also performed a qualitative comparison between the mean streamfunction for these three estimates, noting that the thermodynamic calculation is larger in the lower stratosphere and with more differences between the downward-control calculation and the other two estimates higher up in the stratosphere at the poles.

In order to understand some of the properties of the different vertical velocity calculations, we examine the vertical auto-

correlation of the tropical upwelling velocity for the WACCM model. Note that the "downward control" calculation of $\bar{w}_M^*$ includes the gravity wave drag, since this made an important contribution in the model (but not in the reanalyses, where the gravity wave drag is much smaller).

Figure 2 (panels b-d) shows these autocorrelation coefficients for the three methods with the "tropics" defined as between 30°S and 30°N. The "direct" method is shown in b, and the correlation is broader than the equivalent autocorrelation for the

global diabatic circulation. The "downward control" method is shown in c, and the vertical autocorrelation is even greater. The downward-control method means that upper levels directly impact lower levels (through integration), and so it is consistent that the vertical autocorrelation of $\bar{w}_M^*$ is the broadest of all metrics. Note then that the vertical velocity calculated in this way is essentially a single piece of information for the extent of the stratosphere. Differing variability in the upper and lower branches will be comparatively indistinguishable using such a calculation. Previous results suggest that the upper and lower branches

may be distinguished by this metric for subseasonal variability in winter, however (Abalos et al., 2014). The thermodynamic

vertical velocity in panel d demonstrates that the vertical covariance is not necessarily a result of the flow itself, since vertical correlation is much narrower in this case. Unlike the global diabatic circulation, however, there is little apparent anticorrelation between the upper and lower branches of the circulation. There is an interesting feature in the lower stratosphere for this radiatively determined vertical velocity; beneath 70 hPa, the behavior is much more weakly correlated with upper levels than for the other calculations of vertical velocity. This is consistent with the results of Rosenlof (1995), who speculated that the reason for this low level discrepancy was the relatively simple way the radiative heating was calculated, using the radiative transfer code developed for two dimensional models by Yang et al. (1991) and Olaguer et al. (1992). However, this different behavior in the lowermost stratosphere was also found by Abalos et al. (2015) with the same three complex reanalyses used here, and the result holds with the WACCM model here. These three calculations, often treated as the same, are actually somewhat different, especially with respect to the vertical structure of their variability.

Figure 2 (panels e-g) shows the autocorrelation coefficients for the three methods of calculating the vertical velocity with the "tropics" defined as the average within the turnaround latitudes determined each month from the location where $\bar{w}^*$ switches from upwelling to downwelling. These turnaround latitudes vary from narrower than $30°$ at the lowermost levels to closer to $40°$ at the upper levels (see Abalos et al. 2015 Figure 5 for the mean and climatology of these in the reanalyses). Using the true turnaround latitudes instead of set latitudes for the tropics makes the vertical velocity calculated using all three methods have a narrower extent of the vertical autocorrelation. The implication of this is that a good deal of the difference in variability between levels occurs at the edges of the "pipe", where mixing is playing a role. The difference between the two different edge treatments is greatest in the direct calculation (panel e), where the variability of the vertical velocity in the lower stratosphere and upper stratosphere are no longer positively correlated. The lower stratospheric structure now resembles that of the fixed latitude thermodynamic calculation for all three calculations, with very little relation between the variability beneath 70 hPa and above that level. This suggests that the difference between the thermodynamic calculation and the others is unrelated to the treatment of radiation. As above, however, we can conclude that the three different methods of calculation provide different vertical information.

To examine the relationship between the fixed latitude and turnaround latitude calculations, we show the cross correlation between the two for each calculation method in Figure 4. The y-axis is the turnaround latitude and the x-axis is the fixed latitude calculation. It is evident that the different tropical boundaries matter most for the direct calculation method. The high degree of symmetry in the second and third panel imply that, although small differences were visible in the autocorrelations in Figure 2, the choice of boundary is far less important for the momentum and thermodynamic methods.

In Figure 5, we show the matrix of correlation coefficients ($r$) for each version of the residual circulation vertical velocity with each other version. The top row is the WACCM turnaround latitude calculation; the middle row is the WACCM fixed latitude calculation; and the bottom row is the fixed latitude calculation from one of the reanalyses, ERA-Interim.(Behavior is similar for the other two reanalyses.) The first column shows the correlation between the direct calculation on the y-axis with the downward-control calculation on the x-axis. The second column shows the correlation of the thermodynamic TEM vertical velocity $\bar{w}_Q^*$ with the downward-control calculation $\bar{w}_M^*$. The third column shows $\bar{w}^*$ on the y-axis and $\bar{w}_Q^*$ on the x-axis. These correlation coefficients are demonstrating the interchangeability (or lack thereof) of these different calculations

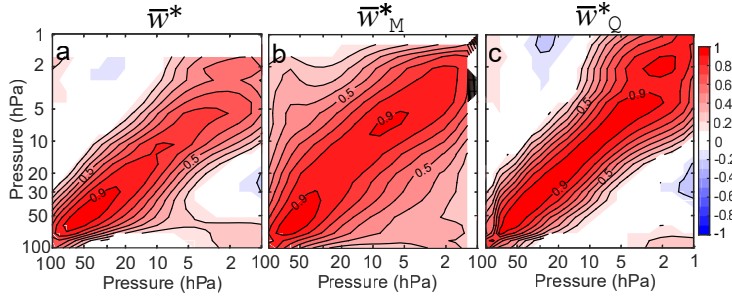

**Figure 4.** Cross-correlation of the variability of the calculations of the residual circulation with the turnaround latitudes (y-axis) and the fixed tropics, 30°S and 30°N, (x-axis). a) is the direct calculation b) is the momentum calculation, and c) is the thermodynamic calculation. Contour lines are every 0.1.

for the vertical velocity. Examining the turnaround latitude calculations (a-c), one notes that the correlation of the downward-control calculation with either of the other calculations is quite weak, never getting above r=0.9. We hypothesize that this is because the vertical integration, which smears out information in the vertical, makes the averaging using turnaround latitudes less clear, since the turnaround latitudes themselves vary with height. The comparison between the direct calculation and

the thermodynamic calculation in Figure 5(c) has much closer agreement than either comparison with the downward-control method. Correlations between the same vertical velocities in WACCM, but now with fixed averaging latitudes (30°S–30°N), are much higher.

Because the calculation for the vertical velocity averaged between turnaround latitudes is less well defined (sometimes there is more than one zero-crossing, for example), and because the fixed latitude calculation is simpler and therefore more common,

we shall default to using the fixed latitude calculation for the most of the rest of this study, though some comparisons with averaging between turnaround latitudes are included as well.

Now we focus on the lower six panels of Figure 5 to see the differences between the methods with the fixed latitude averages and the differences between the model and the reanalysis. In panel (d), the correlation of the two momentum-based calculations at the same level is very high, with the WACCM correlations appearing very similar to the autocorrelations in Figure 2 and

$r > 0.9$ along the diagonal between 50 and 10 hPa for the reanalysis (g). We see the evidence of the broad autocorrelation of the $\bar{w}_M^*$ as the correlations of the lower level $\bar{w}^*$ with the upper levels of $\bar{w}_M^*$ are much higher than the opposite. We note that when the full downward-control calculation—using contours of angular momentum rather than latitude lines—is applied to calculate the $\bar{w}_M^*$ from ERA-Interim, the correlation with $\bar{w}^*$ is actually somewhat worse (r<0.7 along most of the diagonal, not shown), and even lower (r<0.3 along the diagonal) when the correlation is calculated with 6-hourly data rather than monthly (c.f. the

impact of this calculation on the mean in Ming et al. 2016). We speculate that the worse agreement at higher frequencies is related to either small scale torques that are not captured by the momentum budget at these high frequencies or due to the assumption of instantaneous net-zero flow through each pressure surface, which cannot account for short-term storage. In panels (e) and (h), the correlations with the downward-control calculation and the thermodynamic calculation again reach much

deeper along one axis than the other, associated with the broad vertical autocorrelation of the downward control calculation method. Interestingly, at upper levels in the model, this cross-correlation is strongest, while in the reanalysis, the upper levels are where the cross-correlation is weak. The weak correlation at upper levels in the reanalysis product could be a result of the discontinuities in the heating rates above 5 hPa noted by Abalos et al. (2015). The correlation beneath 70 hPa is weak in the

5  model and is not significant in the reanalysis, again consistent with the substantial differences at low levels seen in the mean by both Rosenlof (1995) and Abalos et al. (2015). There are major discrepancies between the lower stratospheric heating rates in different reanalyses, which could explain this feature to some extent (Wright and Fueglistaler, 2013). Panels (f) and (i) show $\bar{w}^*$ on the y-axis and $\bar{w}_Q^*$ on the x-axis. These compare more favorably than the middle column, but it is important to note that even in the WACCM model with these fixed latitudes, these are not equivalent beneath 70 hPa. In the reanalysis, the correlation

10  of these is a bit higher than for the comparison in panel (h), but again there is limited correlation in the upper stratosphere. Because of their different time evolution, it is not entirely surprising that the trends in the circulation calculated using these different methods disagree with each other for the reanalyses (Abalos et al., 2015).

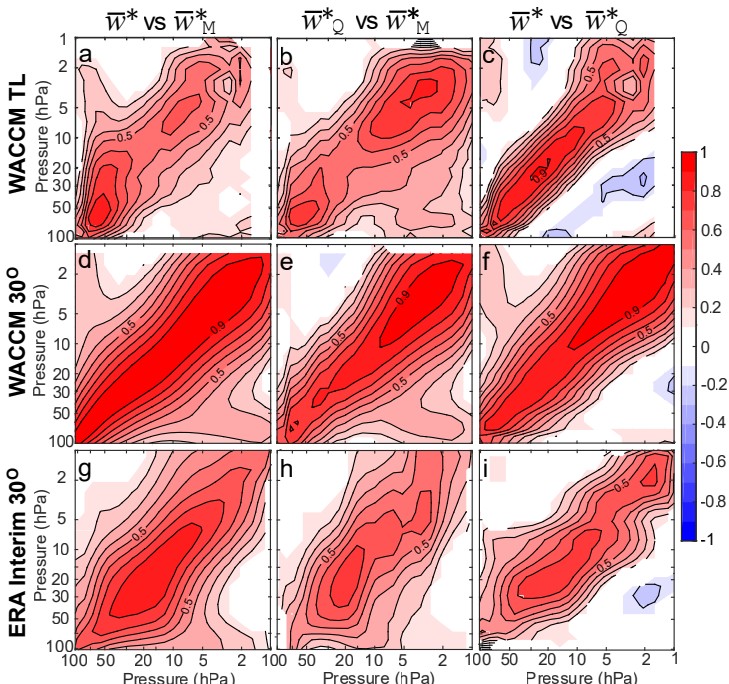

**Figure 5.** Correlation coefficients ($r$) for the deseasonalized time series of the three different TEM vertical velocities calculated from WACCM (a-f) and for ERA-Interim (g-1): (a,d,g) $\bar{w}^*$ vs. $\bar{w}_M^*$, (b,e,h) $\bar{w}_Q^*$ vs. $\bar{w}_M^*$, and (c,f,i) $\bar{w}^*$ vs. $\bar{w}_Q^*$. The quantity plotted on the y-axis is listed first. (a-c) use averaging between the turnaround latitudes while (d-f) are averages between $30^\circ$S–$30^\circ$N. Contours are spaced every 0.1.

## 4.2 TEM vertical velocity compared to the global diabatic circulation

Next, we seek to answer the question of how the global diabatic circulation on isentropes relates to these metrics. We calculate the correlation of the three different calculations of the TEM vertical velocities averaged over 30°S–30°N with the deseasonalized global diabatic circulation on each isentrope (as defined above) for each of the three reanalysis data products and for the WACCM model. For WACCM and for JRA55, we also show these crosscorrelations with the TEM vertical velocities averaged between the turnaround latitudes. (We show only JRA55 because its behavior is very similar to the other two reanalyses. ERA-Interim has slightly higher correlations throughout and MERRA has slightly lower correlations throughout, but the overall patterns are very similar.) These eighteen correlation coefficient matrices are shown in Figure 6.

The highest correlation is found between the global diabatic circulation and $\bar{w}_Q^*$, as expected, because these are both calculated from the heating rates for all three reanalyses and the model. In addition, this comparison has the smallest vertical extent, consistent with the narrower extent of vertical autocorrelations seen in Figure 2. The absolute highest correlations are between the global diabatic circulation and $\bar{w}_Q^*$ averaged between turnaround latitudes in the WACCM model. Interestingly, when comparing the turnaround latitudes to 30°S–30°N for this crosscorrelation in JRA55, the opposite result is seen than for the model. In JRA55 (and for the other two reanalyses, not shown), the correlation between the fixed latitudes is higher. This means that the turnaround latitude averaging introduces more spurious variability in the reanalysis products, while in the model, using the true turnaround latitudes provides closer agreement with the global diabatic circulation. This seems only natural, since the global diabatic circulation is the average of the total mass flux through the surface instantaneously and therefore itself counts for motion of the turnaround latitudes.

For the correlation between the global diabatic circulation and $\bar{w}_Q^*$ in the other three reanalysis calculations, the 50 hPa $\bar{w}_Q^*$ variability is captured in by the 450–500 K global diabatic circulation. The 10 hPa $\bar{w}_Q^*$ variability is captured in all three reanalyses by the global diabatic circulation between 800–900 K. The climatology of the potential temperature–pressure relationship in the tropics (20°N-20°S) is shown in the dashed line. Note that because the diabatic circulation reflects the global circulation while vertical velocities are calculated only in the tropics, the highest correlations are not necessarily expected to be along this line, but it is a useful visual guide. In all three reanalyses and the model, there is some reflection of the anticorrelation of the upper and lower branches of the circulation that is seen in the global diabatic circulation on isentropic surfaces. The relationship with the other TEM vertical velocities is less clear in the reanalyses, though still quite strong in the WACCM model. In the reanalyses, $\bar{w}^*$ at 70 hPa is not strongly correlated with the global diabatic circulation at any level, with the correlation coefficient only reaching up to $r = 0.5$ (at 550 K for both JRA55 and MERRA and between 550 and 650 K for ERA-Interim). The momentum derived vertical velocity is the least well correlated, with the lower level global diabatic circulation having almost no covariability with $\bar{w}_M^*$ at any level except in WACCM. We conclude from this comparison that the global diabatic circulation is very closely related to the TEM vertical velocity calculated using heating rates with less covariation with $\bar{w}^*$ and even less with the momentum derived vertical velocity, $\bar{w}_M^*$. Similar to Abalos et al. (2015), we generally see as much difference amongst the different estimates of the vertical velocity as between the three reanalyses. The WACCM results demonstrate that the tropical upwelling averaged between 30°S–30°N and the global diabatic circulation,

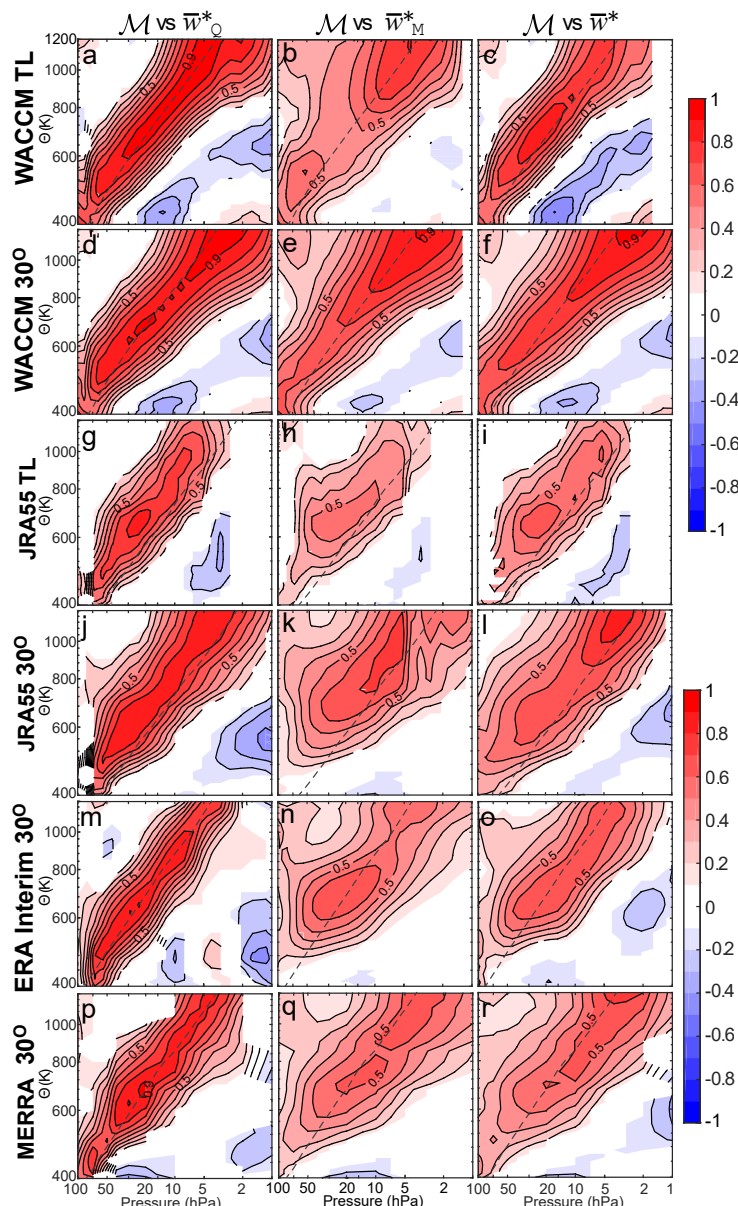

**Figure 6.** The correlation of three different estimates of the TEM vertical velocity with the global diabatic circulation $\mathcal{M}$. The first row is the correlation of the global diabatic circulation with $\bar{w}^*_Q$; the second row is the correlation of the global diabatic circulation with $\bar{w}^*_M$; and the third row is the correlation of the global diabatic circulation with $\bar{w}^*$. The first column shows MERRA, the second column shows JRA55, the third shows ERA-Interim, and the fourth shows WACCM. The gray dashed line shows the climatological relationship between pressure and potential temperature, averaged between 20°S–20°N. All residual circulation velocities are integrated over 30°S–30°N. Contours are spaced every 0.1.

while closely related, are not equivalent. Although the comparison for the thermodynamic vertical velocity with the global diabatic circulation is in places greater than 0.9, the comparison with the other TEM calculations reveals differences, especially lower in the stratosphere. When turnaround latitudes are used instead, the correlations become worse for the global diabatic circulation with both $\bar{w}_M^*$ and $\bar{w}^*$. However, the correlation with $\bar{w}_Q^*$ suggests that these two are very nearly identical, especially above the middle stratosphere. In a model, they could be used interchangeably, but in reanalysis, they are quite different.

## 5  The global diabatic circulation and the water vapor tape recorder

As discussed in the introduction, the water vapor tape recorder can be used to calculate an effective velocity ($w_{TR}$) by tracking the seasonal cycle as it is moved along by the BDC and is another way to get at an "observed" circulation. We modify previous approaches (Niwano et al., 2003; Schoeberl et al., 2008) by using four levels (instead of two) for a phase-lagged correlation. This modification appears to better capture inter-annual variability (e.g., QBO) whereas the two-level method is better at capturing the seasonal cycle (Glanville and Birner, 2017).

We correlate monthly data between three lower levels (z to z+2) and three upper levels (z+1 to z+3) such that the two middle levels overlap. We then calculate the correlation coefficients, shifting the upper level data from +1 to +9 months while holding the lower level still. The lag with the largest correlation coefficient represents the approximate time needed for the tape recorder signal to ascend from the lower levels to the upper levels. The tape recorder speed, assigned to the midpoints between the levels and the time steps, is simply the distance between the levels divided by the time lag. This modified method was tested on various scenarios within a 1-D model and was found to successfully capture variability but underestimate speeds by 5-10%. This method of calculation improves the representation of interannual variability (like the QBO) compared with a simpler two-level method.

It should be noted that methane oxidation acts as a water vapor source, affecting the mean values above 70hPa ($\sim 450$ K), but with smaller impacts on the interannual variability up to about 10hPa (Kawatani et al., 2014). Depending on the seasonal cycle of the methane and the speed of the BDC, this can result in an apparent slow-down, speed-up, or nothing at all. For example, if oxidation occurs before (after) the wet signal, the effective velocity will appear stronger (weaker). However, if oxidation is concurrent with the wet signal, the velocity calculation will not be affected.

Note that although reanalysis products do output water vapor, the inconsistencies of the water vapor tape recorder with the vertical velocities in reanalysis, likely due to enhanced dispersion from the assimilation process, lead us to omit their analysis (Glanville and Birner, 2017). The results of the water vapor tape recorder comparison to the global diabatic circulation are shown for WACCM at 500 K in the time series in Figure 7. This figure shows the significant correlation between the water vapor tape recorder vertical velocity and the global diabatic circulation (r=0.57), and certain features stand out. The QBO appears to be related to a significant fraction of the covariation of these two time series, and when examined, the coherence drops off with periods shorter than the annual timescale. The water vapor tape recorder vertical velocities also appear to have greater decadal variability than the global diabatic circulation. The correlation of these improves upon filtering to remove the higher frequency variability in the global diabatic circulation, which the water vapor tape recorder does not capture. The

correlation between these two measures of the circulation is not strong enough for them to be considered equivalent, in the way that the WACCM results above suggest near equivalency between the global diabatic circulation and the tropical residual circulation vertical velocities for considering interannual variability. $w_{TR}$ results from observations must be understood within this context.

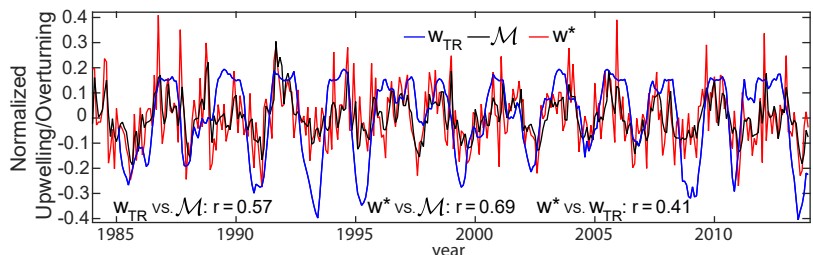

**Figure 7.** Time series of water vapor tape recorder calculated (blue), the global diabatic circulation (black) on 500 K, and $\bar{w}^*$ averaged between $30°$S–$30°$N (red) from the WACCM model. All three timeseries have been deseasonalized and scaled by their standard deviations. Correlation coefficients between each pair of time series are reported at the bottom.

To examine the correlation more broadly, we show the cross correlation between the water vapor tape recorder vertical velocity and the global diabatic circulation at every level in the left panel of Figure 8. The correlation is around 0.5–0.6 along the diagonal, with anticorrelation of up to 0.4–0.5 between the upper branch and lower branch (regardless of the metric). Interestingly, the correlation with the TEM tropical vertical velocity averaged between $30°$S–$30°N$ is considerably weaker, as shown in the middle panel. Note that when the correlation between the TEM tropical vertical velocity and the $w_{TR}$ calculated in pressure coordinates, the magnitude of the correlation is the same as with the isentropic coordinates, except between 5–10 hPa where it is much weaker, not shown. When the TEM tropical vertical velocity is averaged between the true turnaround latitudes, however, the correlation becomes stronger than the correlation with the global diabatic circulation (panel c).

This combination of results—i.e. that the coherence drops off at periods less than a year, that the correlation of the water vapor tape recorder with the global diabatic circulation is stronger than with one type of averaging for $\bar{w}^*$ but weaker than with the other—suggests that the $w_{TR}$ is mostly recording longer timescale variations, and its correlation with the other vertical velocity metrics is mostly to do with which ones respond the same way with the QBO. The anticorrelation seen in Figure 8 (c) is as strong as the correlation along the diagonal. Why the $\bar{w}^*$ averaged between the turnaround latitudes has a response to the QBO that is most similar to that of $w_{TR}$ is unclear. The turnaround latitudes are the narrowest in latitude near the tropopause, where the $w_{TR}$ signal is set, and perhaps this geometry matters. A takeaway from this is that, if one were to compare model results to water vapor observations, none of the dynamical vertical velocity metrics from the model would be appropriate comparisons. Instead, the model's water vapor tape recorder velocity would need to be used. This limits the usefulness of $w_{TR}$ as an observable metric for evaluating reanalyses.

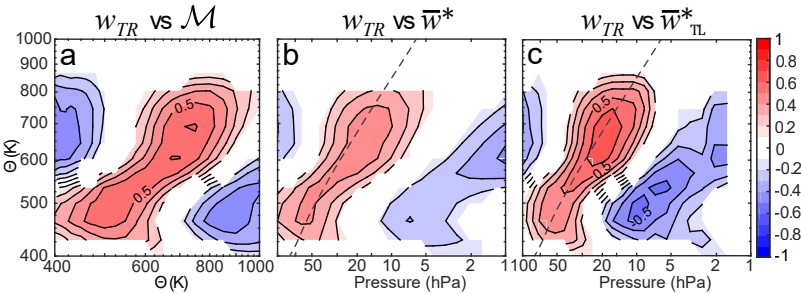

**Figure 8.** Correlation coefficient of the interannual variability of $w_{TR}$ with a) the global diabatic circulation and b) the residual circulation vertical velocity, $\bar{w}^*$ at different levels in the stratosphere. The climatological relationship between pressure and potential temperature (averaged between $20°$S–$20°N$) is shown in b and c in the dashed gray line.

## 6   The global diabatic circulation's relationship with ozone

One motivation for studying the BDC is its influence on radiatively important trace gases, such as water vapor and ozone. Water vapor is a quasi-conserved tracer once it enters the stratosphere (in the absence of the aforementioned methane oxidation), and so its behavior is comparatively straightforward. In contrast, ozone is both produced and destroyed in the stratosphere in chemical processes that are photochemically and temperature dependent. The ozone maximum is around 7 hPa or 800 K in the tropics (e.g. Paul et al., 1998), where photolysis by wavelengths less than 240 nm dissociates molecular oxygen (Chapman, 1930; Seinfeld and Pandis, 2006). As air moves from the tropics, it advects the ozone to mid and high latitudes. Stratospheric ozone absorbs ultraviolet radiation, creating heat, and thereby it influences the thermal structure of the stratosphere (e.g. Andrews et al., 1987) and thus the diabatic heating and transport. As the chemistry itself is temperature dependent, ozone, temperature and the circulation are closely connected.

With this interconnectivity in mind, we examine the total column ozone correlation at every latitude with the the global overturning circulation at each level within the stratosphere. The correlation of the deseasonalized time series of the monthly mean total column ozone data from the Solar Backscatter Ultraviolet Instrument (SBUV) from 1980–2013 and the global diabatic circulation from the three different reanalyses is shown in Figure 9. Also shown is the correlation of the total column ozone and global diabatic circulation from the WACCM model. Generally, there is a consistent pattern across all three reanalyses and the model. This pattern is consistent with the ozone variability associated with the QBO: an out of phase relationship between the lower and upper stratosphere and an out of phase equatorial and subtropical pattern (e.g. Zawodny and McCormick 1991). The ERA-Interim correlation with the SBUV data is much stronger than the correlations of the other two reanalyses with the SBUV data. Note that ERA-Interim assimilates the SBUV data, where MERRA and JRA55 do not, and this is a likely explanation for the increased correlation. Nevertheless, as the same spatial patterns are visible in the correlations with all three reanalyses and the model, we consider them to be robust and seek to understand them–i.e. whether they are due almost entirely to the QBO as with water vapor, or whether other dynamical variability is important. We will focus on WACCM, as its dynamics are necessarily consistent with the ozone concentrations.

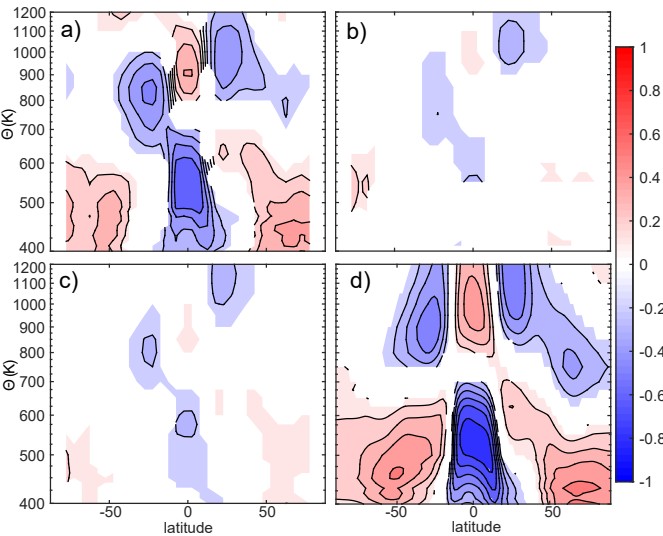

**Figure 9.** Correlation coefficient ($r$) of the interannual variability of total column ozone at every latitude to the total diabatic overturning circulation at every level for (a) ERA-Interim $\mathcal{M}$ and SBUV total column ozone, (b) JRA55 $\mathcal{M}$ and SBUV total column ozone, (c) MERRA $\mathcal{M}$ and SBUV total column ozone and (d) ozone and $\mathcal{M}$ from WACCM.

We see that the high latitude total column ozone is correlated with the circulation in the lowermost stratosphere, with the correlation explaining up to 25% of the deseasonalized total column ozone variability in the Northern hemisphere polar region. The total column ozone in the tropics is strongly anticorrelated with the global diabatic circulation around 500 K. Both of these are qualitatively consistent with transport being the dominant factor driving the relationship between the variability in ozone and in the circulation at these levels. The correlation is strongest in the Southern hemisphere in the collar region of the polar vortex, around 55°S, and is weaker at the pole, while in the Northern hemisphere, the correlation is stronger poleward of that, around 70°N. More air is transported by the global diabatic circulation and mixing to the Northern hemisphere pole than the Southern hemisphere pole because the Southern hemisphere polar vortex is a stronger barrier to mixing. The tropical total column ozone is also correlated with the circulation at upper levels, above the ozone maximum (800 K). Like water vapor, the correlation is related to the QBO, and is strongest at 2 year periods (not shown). Some coherence at higher frequencies can be explained through the anticorrelation of the upper and lower branches of the circulation discussed above. The subtropical total column ozone is anticorrelated with the upper level circulation strength, with hemispheric asymmetry in which levels relate to the subtropical ozone in the different hemispheres. This is consistent with previous results showing the meridional pattern associated with the QBO at these levels leads to opposite anomalies in the deep tropics and the subtropics (e.g. Randel et al., 1999; Tian et al., 2006).

To examine these correlations further, we plot the correlations of the deseasonalized ozone concentrations at each latitude and pressure from WACCM and the deseasonalized global diabatic circulation at two individual levels in Figure 10. In this way, we try to understand where in the stratosphere the total column ozone correlation patterns are determined. The top left

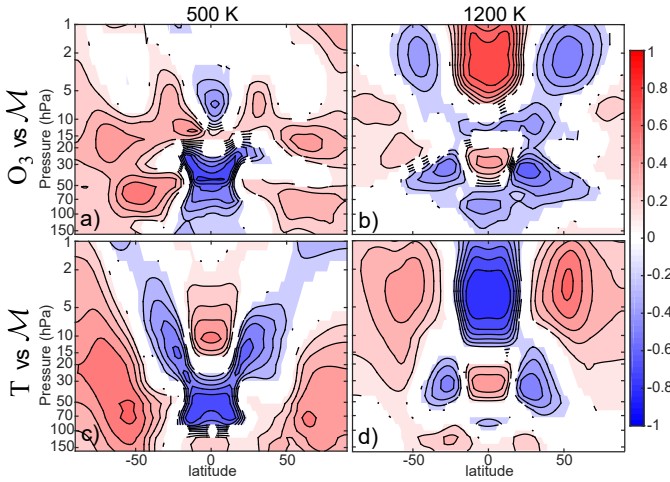

**Figure 10.** Correlation coefficient ($r$) of the interannual variability of local ozone concentration at every latitude and pressure to the global diabatic circulation at (a) 500 K and (b) 1200 K from WACCM. Correlation coefficient ($r$) of the interannual variability of local temperatures at every latitude and pressure to the global diabatic circulation at (c) 500 K and (d) 1200 K from WACCM. Contours are every 0.1, and correlations are only plotted where they are significant at the 95% confidence level.

panel shows the correlation of the local ozone concentration with the global diabatic circulation at 500 K. The strong signal beneath the ozone maximum is consistent with the transport driving the ozone variability—upwelling ozone poor air from the troposphere and exporting the high ozone tropical air to the midlatitudes and poles in both hemispheres. The top right panel shows the correlation of the deseasonalized local ozone concentration with the global diabatic circulation at 1200 K. At the

5 equator at upper levels, the correlation is high, and the strong subtropical signal we see in Figure 9 is related to the variability of ozone at the uppermost levels and the local ozone concentration on the edge of the tropics in the lower branch. As has been previously reported (Perliski et al., 1989), there is a division between what drives ozone variations in the upper and lower stratosphere. Our results for the global diabatic circulation are consistent with two different processes being responsible for these differing behaviors: Near, at, and above the ozone maximum, the ozone distribution is determined by chemistry, while at

10 the lower levels the ozone distribution is determined by transport. Evidence of these two separate processes is discussed below.

The correlation of the upper level circulation with the lower level ozone concentrations on the edges of the tropics is consistent with the anticorrelation of the upper and lower branches of the circulation and different characteristics of the transport. In the lower branch, the stratospheric entry latitudes are close to the poleward flanks of the tropics (Birner and Bönisch, 2011), and so if the anticorrelation of the upper and lower branches of the circulation is a partitioning between the deep tropical entry

latitudes and the more subtropical entry latitudes, the strong upper branch is associated with less upwelling in these flanks and thus less ozone around these turnaround latitudes. This hypothesis of the partitioning of the circulation between upper and lower branches at monthly to seasonal periods and its relationship with trace gas transport is the subject of further study.

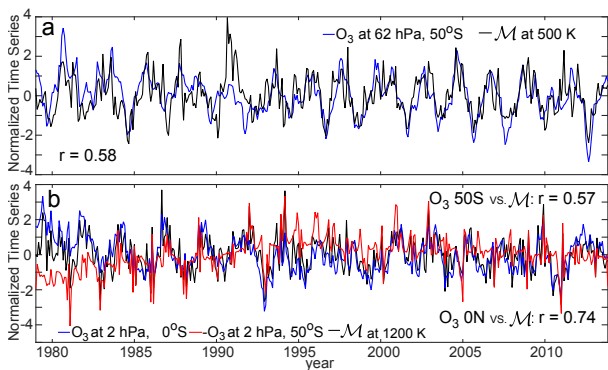

**Figure 11.** Timeseries of monthly mean local ozone concentration and the global diabatic circulation from WACCM for (a) 62 hPa, $50°$S $O_3$ in blue and $\mathcal{M}$ at 500 K in black and (b) 2 hPa equatorial $O_3$ in blue, 2 hPa, $50°$S $O_3$ in red (multiplied by -1), and $\mathcal{M}$ in black.

Figure 11 shows time series of the local ozone concentrations and total overturning strength based on the maximum correlations shown in Figure 10. Figure 11 (a) shows the tight coupling between the ozone in the Southern hemisphere midlatitudes with the global overturning strength at 500 K. Figure 11 (b) shows the very close correlation of the upper level circulation and the upper level equatorial ozone and the weaker negative relationship with the upper level midlatitude ozone. The two timeseries in (a) and the equatorial ozone and global overturning in (b) are correlated at all timescales, while the anticorrelation between the the midlatitude ozone and the upper level circulation strength is stronger at short timescales. Ozone variability at upper levels is dominated by photochemical processes (Perliski et al., 1989), resulting in a short chemical lifetime, and this close correlation is due to the relationship of temperature with both ozone and the circulation strength. When the circulation is stronger in the tropics at these levels, that is associated with cooling and consequently longer ozone chemical lifetimes. We have therefore plotted the correlation of the temperature with the global diabatic circulation at 500 K and 1200 K in Figure 10 (c) and (d). In both (c) and (d), it is evident that at low levels the temperature and ozone respond to the circulation similarly. In (d) in particular, the opposite relationship between the circulation and the temperature is observed to the circulation and the ozone, which indicates that the temperature is driving the chemistry at upper levels. To test this mechanism, we have plotted the natural log of the ozone concentrations against the inverse of temperature at these upper levels and at lower levels, since an exponential dependence on the inverse of temperature is a form that is consistent with the form for many of the reaction rate coefficients for ozone loss processes (Stolarski et al., 2012). These results are shown in Figure 12. Clearly, the upper level and lower level are behaving differently: because it is dynamically controlled, the lower level ozone depends as much on latitude as on the inverse of temperature; the slope is determined by the relative vertical gradients of temperature and ozone. The upper level has little latitudinal dependence and a positive slope, consistent with the chemical control. When the fit is calculated for 45-50$°$S at 1 hPa, as shown in the third panel of Figure 12, the slope agrees to within error with the slope calculated for the Limb Infrared Monitor of the Stratosphere (LIMS) data used by Stolarski et al. (2012). We have taken the opportunity to show the change in the relationship over time using different colors. Calculating the fit for just the earlier years results in a higher value for the "initial" ozone concentration with a slope that is the same to within error. While we do not investigate the cause

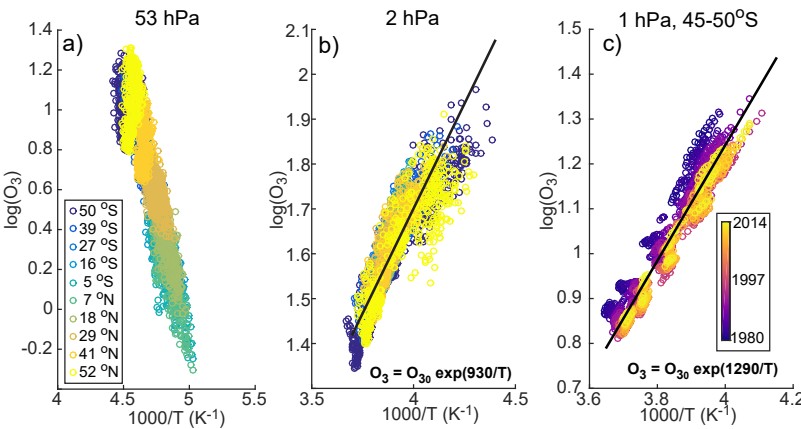

**Figure 12.** The natural log of the ozone volume mixing ratio (in ppm) and the colocated values of 1000/T for different levels and latitudes in the WACCM model for 1980–2014. a) is at 53 hPa and b) is at 2 hPa, both for all latitudes equatorward of 52°. c) is at 1 hPa for 45–50°S only. In a) and b), different colored circles are different latitudes, with the Northern Hemisphere being yellow and the Southern Hemisphere blue. In c), different colors show different years. In b) and c) the best fit lines are also plotted in black with correlation coefficients of $r = 0.87$ and $r = 0.96$ respectively.

for this change here, we hypothesize that it is related to the higher mean ozone concentrations being advected to this region during the initial period of the ozone hole.

Stratospheric transport timescales for even the lower branch are around half a year to a year (Orbe et al., 2014), and so instantaneous correlation plots, as in Figure 9, might seem to be less relevant. The global diabatic circulation necessarily
integrates over that transit time, however, as it accounts both for variability in the upwelling region and in the downwelling region simultaneously. Therefore we do not perform lagged regressions to attempt to understand causality. Rather we suggest that the use of frequency dependent correlations, which will have a corresponding phase lag (e.g. Swanson, 2000), would be necessary to look at the causal relationships. However, as we can see from Figure 11, the correlations are in phase at monthly timescales, and so higher frequency records (minimum daily) will be necessary to diagnose the phase (and thus the implied
causality) in these relationships.

The correlations between ozone and the global diabatic circulation have resemblance to the pattern of the ozone response to the QBO, but coherence at higher frequencies suggests that other processes play a role, in contrast to the water vapor tape recorder examined above. We have demonstrated the close dependence of ozone variability on the global diabatic circulation variability with ozone data and with reanalysis and model data. The total column ozone at the poles and in the tropics is
correlated with transport by the global diabatic circulation in the lower stratosphere. The results of the investigation of the correlation of ozone with the global diabatic circulation have demonstrated consistency with our understanding of the roles of circulation and chemistry, and so we suggest that the global diabatic circulation can be adopted in this context with little-to-no change in interpretation compared to $\bar{w}^*$.

# 7 Discussion and Conclusions

In this study, we have compared the global diabatic circulation to the more typically used metrics for the strength of the BDC and to tracers. In particular, we have examined the residual circulation vertical velocity, the water vapor tape recorder, and total column ozone concentrations.

We find that the three common methods for quantifying the BDC strength from models and reanalysis data products have somewhat different deseasonalized variability, especially in the lower stratosphere. We also find that the choice of averaging latitudes—whether fixed tropics (30°S-30°N) or turnaround latitudes—has an effect on the deseasonalized variability that depends on the method. These methods also result in different vertical structures; the calculation based on the principle of downward control has a much broader vertical autocorrelation than the calculation from diabatic heating rates. The direct method is somewhere in between, and when the turnaround latitudes are used, it becomes very similar to the calculation from diabatic heating rates. Thus, if the separate evolution of the upper and lower branches of the circulation are of interest, the most appropriate metric is one that uses the diabatic heating rates or the direct method with turnaround latitudes. In the model, the relationship of the different TEM calculation methods with fixed tropics are nearly one-to-one above 70 hPa. For the reanalysis products, however, the differences between calculations of the TEM $\bar{w}^*$ are quite distinct, especially at lower levels where they are often analyzed. The comparison between the TEM $\bar{w}_Q^*$ with the diabatic overturning circulation is as favorable as the comparison between the TEM $\bar{w}_Q^*$ and the $\bar{w}^*$ from the residual circulation method. In general, consistency between methods is better higher up in the stratosphere, while beneath 70 hPa, the differences between the methods is substantial. These results suggest that the method of calculation could significantly affect comparisons between the residual circulation from reanalysis and any other observed stratospheric variable.

Like the thermodynamically constrained $\bar{w}_Q^*$, which the global diabatic circulation so closely resembles, the global diabatic circulation requires only monthly mean heating rates, temperatures, and pressures. Its calculation is simpler than that of $\bar{w}*_Q$, which requires some assumption about how to enforce mass conservation (Abalos et al., 2012), and which can have complications with convergence when the iterative solving method converges but then occasionally proceeds to diverge after additional iterations. Eddy terms in the thermodynamic equation are neglected in the calculation, which may be a reason for these convergence difficulties. The global diabatic circulation also has an interesting property that the lower and upper branches of the circulation are anticorrelated, so that when the lower branch is stronger, less air is flowing through the upper branch. This is even more curious when one takes into account that the vertical velocity in the lower stratosphere is the sum of both branches. This pattern might be expected with the QBO, but as the coherence is not just at QBO frequencies, an additional mechanism is necessary. One explanation is that this could be due to a change in index of refraction when there is more total wave activity that causes higher amplitude planetary scale waves to break lower in the stratosphere; we have yet to test this mechanism. Another possibility is that the meridional location of the wave breaking changes such that when the lower branch is stronger, less wave activity can propagate up into the upper stratosphere. Alternatively, there may be an interaction between planetary and gravity waves. The anticorrelation is consistent with the conclusions of both Ray et al. (2010) and Stiller et al. (2012), who concluded based on observations that the trends in data were best explained by a strengthening in the lower branch of the

circulation and a weakening in the middle and upper stratosphere. The ERA-Interim trends in the global diabatic overturning circulation are consistent with this picture, although the upper-level trends are problematic because of the changes in observing systems. The other two reanalyses do not agree.

The global diabatic circulation is correlated with the water vapor tape recorder vertical velocity, especially at intraseasonal and longer time scales. Perhaps unsurprisingly, given its close theoretical relationship with stratospheric tracers, the global diabatic circulation is a predictor for the water vapor tape recorder. However, the overall weakness of the correlation, which explains at most <40% of the variability even when both metrics are derived from a model, suggests the inadequacy of using the water vapor tape recorder as a lone observational record of the changing stratospheric circulation. Rather, the water vapor signal should be compared to water vapor in models in order to assess the combined effect of diabatic heating, diabatic diffusion, and adiabatic mixing.

We analyze the impact of the global diabatic circulation on total column ozone using satellite data and the three reanalyses, including examining the dependence of the total column ozone on different vertical levels of the circulation. When we find consistent behavior amongst the three reanalyses, we explore the mechanism using a model which shows the same behavior. We find that the tropical ozone is most correlated with the overturning at 500–550 K, the Southern hemisphere ozone is sensitive to the overturning at around 480 K, and the Northern hemisphere ozone is most sensitive to the overturning at 400–450 K. The subtropics are most sensitive to the midlevel circulation at 800–1000 K, related to dominant role of chemistry at upper levels. Generally, the patterns associated with the ozone correlation with the global diabatic circulation are consistent with much of this relationship being related to the QBO.

Based on its close relationship with one of the common metrics for the BDC, the ease of calculation, the demonstrated impact on ozone and water vapor, and the constraints provided by tracer observations, we present the global diabatic circulation as a metric for the BDC that should be newly considered. Before the community settled on $\bar{w}^*$, the global diabatic circulation was used (Pyle and Rogers, 1980; Rosenfield et al., 1987). Some intuition for the behavior of $\bar{w}^*$ exists, but both Abalos et al. (2015) and this work have demonstrated that the various methods of calculation are not equivalent, especially for renalyses. Thus, although some variety of TEM $\bar{w}^*$ is the most common metric at present, its calculation is not held in common amongst different studies. In order to understand models and reanalyses, consistency is critical. For the purposes of reanalysis evaluation, therefore, we advocate using the global diabatic circulation along with a version of the quasigeostrophic TEM $\bar{w}^*$ with fixed tropical averaging latitudes (as the turnaround latitudes for the reanalyses are not always well defined, which limits the vertical extent of comparisons). These two metrics rely on different assumptions, and the heating rates from reanalysis might be suspect. For the purposes of model evaluation, the global diabatic circulation should be sufficient. The latitudinal structure of the circulation cannot be examined using the global diabatic circulation, however, and so the vertical velocity should be used when meridional structure is of interest.

Apart from the brief analysis with ozone, this paper does not directly address causality. It is an investigation of different metrics for the circulation from an empirical perspective, revealing that the significant differences in the behavior of these metrics raises questions about their interchangeability, especially for reanalyses. The inconsistencies reveal the extent to which the reanalyses momentum and energy budgets are not internally consistent. At upper levels, the different vertical velocities are

all nearly equivalent, but at lower levels, and especially beneath 70 hPa, the differences are substantial. In particular, using the momentum-based calculation for the residual circulation vertical velocity will mask variability that is not coincident between the upper and lower branches, while the global diabatic circulation emphasizes the difference between the upper and lower branch. This work serves as motivation for additional, process-based and theoretical studies that address the causes of these
differences between the residual circulation metrics and between tracers and the residual circulation.

*Competing interests.*  We declare that we have no competing interests.

*Acknowledgements.*  We thank R.A. Plumb, D. Ivy, and S. Solomon for helpful discussions. This research was conducted with government support for ML under and awarded by the DoD, Air Force Office of Scientific Research, National Defense Science and Engineering Graduate (NDSEG) Fellowship, 32 CFR 168a. Additional funding for ML was provided by NSF award AGS-1608775. This work was supported in part
by the National Science Foundation grant AGS-1547733 to MIT. MA was supported by funding from the Program Atracción de Talento de la Comunidad de Madrid (2016-T2/AMB-1405) and the Spanish National Project STEADY (CGL2017-83198-R). AM acknowledges funding support from the European Research Council through the ACCI project (Grant 267760) lead by J. Pyle. The National Center for Atmospheric Research (NCAR) is sponsored by the US National Science Foundation. Any opinions, findings, and conclusions or recommendations expressed in the publication are those of the author(s) and do not necessarily reflect the views of the National Science Foundation. WACCM
is a component of the Community Earth System Model (CESM), which is supported by the National Science Foundation (NSF) and the Office of Science of the US Department of Energy. Computing resources were provided by NCAR's Climate Simulation Laboratory, sponsored by NSF and other agencies. This research was enabled by the computational and storage resources of NCAR's Computational and Information System Laboratory (CISL). Part of this research was carried out at the Jet Propulsion Laboratory, California Institute of Technology, under a contract with the NASA Aeronautics and Space Administration. One of the datasets used for this study is from the Japanese 55-year
Reanalysis (JRA55) project carried out by the Japan Meteorological Agency (JMA). MERRA was developed by the Global Modeling and Assimilation Office and supported by the NASA Modeling, Analysis and Prediction Program. Source data files can be acquired from the Goddard Earth Science Data Information Services Center (GES DISC). ERA-Interim data provided courtesy of ECMWF. SBUV data courtesy of NASA and are available at

https://acd-ext.gsfc.nasa.gov/Data_services/merged/

.

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
