# Peer review of "The global diabatic circulation of the stratosphere as a metric for the Brewer-Dobson Circulation"

_Atmospheric Chemistry and Physics, 2018_

## Referee Comment (RC1) · S. Dietmüller (Referee) · 15 Nov 2018

**Interactive comment on "The global overturning diabatic circulation of the stratosphere as a metric for the Brewer-Dobson Circulation" by Linz et al. 2018**

This study uses the recently introduced new metric of the BDC, the global overturning diabatic circulation, and investigates its relationship to more traditionally used metrics of calculating the BDC (i.e. different TEM residual circulation metrics). The authors show this relationship for a state of the art climate chemistry model (WACCM) and also for three different reanalysis data sets (ERA-Interim, MERRA, JRA-55). Comparing the different TEM metrics to the global overturning diabatic circulation, they found mainly

good agreement in the middle and higher stratosphere, while in the lower stratosphere the difference between the methods is substantial (with highest differences in the reanalysis products). Moreover this paper includes a very nice analysis about the correlation of the diabatic circulation with water vapor tape recorder and also with total column ozone. The results are well organized and described and the topic is appropriate certainly of interest for ACP. I recommend publication with consideration of the specific comments below.

**Specific comments**

Pg. 2, line 18: '... transport processes such as mixing'. Please include some literature here.

Pg. 2, line 25: You use different terms for 'global overturning diabatic circulation' in the text, e.g. total overturning circulation, total global diabatic circulation, diabatic overturning circulation, diabatic circulation, global average overturning circulation. Perhaps it is easier for the reader to use the same term in the entire text.

Pg. 4, line 4: What is the horizontal resolution of the model?

Pg. 4, line 26: I do not understand how the QBO influences the correlations between diabatic circulation and other TEM calculations? As I understand, the QBO should have the same influence on the interannual variability of all BDC metrics, and thus it should not influence the correlations, or?

Pg. 5, line 1: Tropical Leaky Pipe *Model*: Perhaps you could shortly explain the TLP model.

Pg. 5, line 6: It would be nice to have an additional sentence about the advantage that the global mean overturning circulation can be assessed from observational data (so you can refer to the statement made at Pg. 4, line 32).

Pg. 7, line 9: Reword to '... primary diagnostic of the stratosphere for models and observations'. And what do you mean with TEM diagnostic from observations?

[Figure]

Pg. 8, line 6: Can you explain why you use 30N/S as latitude band? Do you know how sensitive the calculations are if you vary the latitude band to 20N/S or to turn around latitudes?

Pg. 8, line 22: Do you know the reason why it is important for the model and not for reanalysis?

Pg. 8, line 35: Could you mention how the radiative heating was calculated in Rosenlof (1995), e.g. with a radiative transfer model?

Pg. 9, line 16: Could you explain, why the correlation is worse when calculated with higher frequency data? What does the study of Ming et al. say about this issue?

Pg. 12, line 2: It would be easier for the reader, if you could repeat the time period for which the trends are calculated (i.e 1980-2014) here? It was only mentioned in section 2.

Pg. 12, line 4: Abalos et al. 2015 do not look exactly at the same time period (1979-2012) when looking at trends. Moreover QBO and ENSO variability were removed in Abalos et al. 2015. Is it possible that these facts could also explain the mentioned differences in the trends?

Pg. 12, line 9: You mention, that isentropic levels are changing their location over these decades. Did you check this for the data you are using?

Pg. 14, figure 6: Why was the correlation only done with w*, and not with the other TEM residual circulation metrics?

Pg. 15, line 13: Can you explain, why you do not use ozone concentrations from the reanalysis? Correlations of reanalysis data (shown in Fig. 7a-c) would perhaps become better, when the data are more consistent.

Pg. 16, line 2: Perhaps you could add a sentence about, how correlation of ozone with the TEM calculations do look like? Or is their a reason why you didn't look at these correlations?

Pg. 16, figure 7: I am not sure, but perhaps an additional plot of the vertical profile of the diabatic circulation and the latitudinal distribution of the O3 column would be nice, to have an idea how they look like.

Pg. 17, line 17: Could you define stratospheric entry levels?

Pg. 17, line 30: Can you explain why cooling leads to more ozone production? (Or is that clear to everyone?)

Pg.18, figure 9: Please give the correlation value within the plot (as done in figure 5)?

Pg. 20, line 3: '... *metrics for the strength of the BDC*. In particular, we have examined ...... and the total column ozone concentrations.' Ozone column is not a metric for strength of BDC. Please reword the sentence.

Pg. 20, line 19: '...., which can have complication with convergence.'. Can you explain?

Pg. 20, line 30: '... in observing systems.' Some more explanation would be nice, or give a relevant citation.

**Technical corrections**

Pg. 2, line 1: Change 'gases' to 'trace gases'

Pg. 2, line 4: 'surface circulation' – Do you mean surface climate?

Pg. 2, line 11: Perhaps you can reword this sentence, I did have problems to understand it.

Pg. 2, line 20: Change to 'age of air'

Pg. 2, line 26: Perhaps change to '*stratospheric* circulation strength'

Pg. 2, line 31: 'TTL' – tropical tropopause layer (TTL)

Pg. 2, line 33: '10S-10N'

Pg. 3, line 5: *stratospheric* circulation

Pg. 3, line 6: Remove 'it'

Pg. 3, line 15: *stratospheric* circulation

Pg. 4, line 5: Spelling: 'prescribed *observed* sea surface temperature'

Pg. 4, line 7: Change to 'model *simulation*'.

Pg. 4, line 17: Change to 'heating *rates*'.

Pg. 4, line 30: 'circulation *on* isentropes'. There was one 'on' too much.

Pg. 5, line 1: Perhaps change to 'age *of air* tracer'

Pg. 6, line 2: Change to '...is *outputted* differently'.

Pg. 7, line 19: Change to '*zonal mean wind*'

Pg. 8, line 7: Change to '...at *all* levels.'

Pg. 9, line 2: 'Abalos et al.:' the year of the citation is missing

Pg. 9, line 6: Replace 'plots' with *panels*.

Pg. 9, line 21: Change to '...*is* weak'.

Pg. 12, line 4: 'ERA-*Interim*'

Pg. 12, line 8: Change to '...for the vertical *residual* velocity'

Pg. 15, line 3: Change to '*before mentioned*'

Pg. 17, line 2: Change to '...couple *of* individual ...'

Pg. 17, line 24: Perhaps change to *Figure 9(a)* shows ...' This is optical easier to read.

Pg. 17, line 25: Perhaps include ' ....weaker *negative* relationship'

Pg. 19, line 8: Change to '... with *reanalysis and model data*'

Pg. 20, line 6: Remove one 'on' in '....based on the ...'

---

## Referee Comment (RC2) · Anonymous Referee #2 · 15 Nov 2018

The goal of this paper is to improve understanding of the global stratospheric diabatic circulation through isentropes (M) as a metric for the Brewer-Dobson Circulation, by making comparisons to other more commonly-used metrics, including derived tropical upwelling and circulations estimated from water vapor and ozone. The calculation and use of M has certain theoretical advantages to diagnose stratospheric transport in an integrated manner, and it is a good idea to make systematic comparisons to other BDC metrics that are commonly used in the research community. These comparisons can pave the wave for more widespread use of M as a diagnostic tool, as proposed in this paper. I especially like the combination of analyzing reanalysis data sets in tandem with results from a self-consistent chemistry-climate model. This paper will make a valuable contribution and the topic is appropriate for ACP.

While I strongly endorse the goals of the paper, I have a few major comments on the current content, where I think the paper could be improved prior to publication:

1) The current paper focuses on interannual variability in all of the circulation diagnostics. While this is certainly interesting, I suggest also including comparisons of the actual seasonal cycles in various quantities (climatological monthly time series at a few different theta/pressure levels), which can then serve as a context and background for evaluating interannual variability. In order to enhance the understanding of M, it might be useful to include some simple, approximate conversions of the mass flux to equivalent upwelling velocity, to facilitate direct comparisons to the various estimates of tropical upwelling ($w^*$, $w^*_m$, $w^*_Q$, $w_{TR}$). I expect there will be reasonable overall agreement (with, e.g., a large annual cycle in the lower stratosphere).

2) Most of the interannual variability in the results is obviously due to the quasi-biennial oscillation (QBO); this is clearly seen in the time series in Figs. 5 and 9, and the ozone results in Figs. 7-8. This understanding should be folded into the discussions on comparing the behavior of M and various circulation statistics. For example, the vertical out-of-phase behavior between the lower and upper stratosphere is closely tied to the QBO vertical structure. The patterns of ozone variability (out-of-phase in altitude and latitude) and coupling to meridional circulation are well-known aspects of the QBO signal in ozone (e.g. Bowman, 1989, JAS; Zawodny and McCormick, 1991, GRL; Chipperfield et al, 1994, GRL; Randel et al, 1999, JAS; Tian et al, 2006, JGR). Also, the in-phase vs. out-of-phase ozone-temperature relationships in the lower and upper stratosphere, respectively, are a well-known general result tied to transport and photochemistry. While the M comparisons with the various tropical upwelling estimates were novel and interesting to me, I found the results on ozone (Section 6) to be less valuable for evaluating M as a circulation diagnostic (more of a consistency check with previous results).

Minor comments:

1) In addition to the auto- and cross-correlation diagnostics (Figs. 1-3), it would be valuable to explicitly compare time series of the interannual anomalies in all of the circulation diagnostics, like those included for M and $w_{TR}$ in Fig. 5 (perhaps for time series in the lower and upper stratosphere). This very much helps the reader understand the variability that is

quantified in the correlation diagnostics (and provides a 'feel' for the variability among the different diagnostics). Are these comparisons sensitive to the choice of latitude band for the various w quantities?

2) P. 5, line 32: you might include a reference to Abalos et al, 2017, JAS, in regards to trend sensitivity to a tropopause-based vertical coordinate.

3) It would be good to add a few contour labels to the panels in Figs. 1-3 and 6.

4) P.8, lines 25-28: the 'downward control' calculations integrate the wave driving multiplied by density, so that in practice the forcing is usually dominated by nearby levels in the vertical (not the entire depth of the stratosphere).

5) You might note that $w^*_Q$ calculations near the tropopause have an uncertainty in the calculations due to neglect of eddy transport terms (Abalos et al, 2012, ACP). Is this what is meant by 'complications with convergence' (p. 20, line 19)?

6) The dashed lines relating potential temperature and pressure levels in Figs. 3 and 6 are calculated for an ideal gas, and I guess you mean an isothermal ideal gas. Why not just use a relationship derived from climatological mean values, including realistic temperature structure?

7) I was surprised to see no significant trends in the WACCM diabatic circulation in Fig. 4, given that many models (including WACCM) show small positive trends in tropical upwelling (e.g. Garcia and Randel, 2008, JAS). What do trends in the various WACCM $w^*$ quantities look like? If these are different from the WACCM M trends, why is that? Is the QBO variability accounted for in these trend calculations?

8) I do not understand the overlapping 3-level correlation calculations used to derive $w_{TR}$ from the WACCM water vapor fields. Why is such a complicated calculation necessary? How sensitive are the results to different methods of calculation? How does the background annual cycle of $w_{TR}$ compare with the other upwelling estimates (see major comment 1 above).

9) P. 18, line 4: variations in ozone and (potential) temperature are positively correlated in the lower stratosphere because of similarly signed vertical gradients (and long ozone lifetimes), not because of ozone production.

10) P. 20, line 17 and 19: do you mean $w^*_Q$ instead of $w^*$?

11) P. 20, lines 19-30: as noted above, the vertical anti-correlation of the interannual circulation diagnosed here is mainly attributable to the QBO vertical structure (linked to tropical wave dynamics and mean flow interactions). This important aspect should be incorporated into the interpretation and summary discussions of vertical structure.

---

## Referee Comment (RC3) · Anonymous Referee #3 · 19 Nov 2018

Linz et al. present a comparison of the global overturning diabatic circulation with a range of other metrics used to assess stratospheric circulation. They do this using both modelled and reanalysis data. The analysis and discussion presented in the paper is of a high standard and explores an important and relevant topic within the scope of SCP, and as such merits publication following revision. I have several comments the authors should address before publication:

General Comments:

1. The authors use a large number of terms to refer to stratospheric circulation in general and the global overturning diabatic circulation in particular. I feel it would aid

the reader if consistent terms were used throughout the manuscript.

2. Care should be taken when discussing the effects of transport on the distribution of ozone in both sections 1 and 6. It is misleading to say that ozone is produced in the tropics and moved to high latitudes by stratospheric circulation. Brewer and Wilson (1968) and more recently Grewe (2006) highlight that while chemical ozone production in the tropics is high, so is chemical destruction. Grewe (2006) conclude that the view of the tropical region as the global source for stratospheric ozone is highly questionable and that while the tropics tribute to extra-tropical stratospheric ozone, of far greater importance is the production of ozone in the extra-tropics.

3. I miss in the introduction any discussion on the drivers of BDC change or the feedbacks between stratospheric transport and chemical tracers. For example, recent model studies have shown that both GHG increases and polar ozone depletion accelerate the BDC, while polar ozone recovery may to some extent offset an acceleration of the BDC expected from future GHG increases. Of particular importance to this study, these processes have been shown to affect different branches of the BDC (e.g. Braesicke et al., 2014). Some discussion on how these processes change both the speed and morphology of the BDC may aide in interpretation of the correlations presented in the manuscript. Additionally, with a focus on the ozone section, changes to the BDC will alter the distribution of radical source gases, in turn altering stratospheric ozone, which will in turn alter the dynamics. Highlighting the complexity of the coupled dynamical-chemical system and elaborating on how these feed backs operate would in my view strengthen the introduction and prepare the way for the discussion that follows.

Specific comments:

P1L7: The authors could state here which reanalyses and model is used.

P1L14: insert space between 500 and K.

P2L4: perhaps change 'surface circulation' to 'tropospheric circulation' or 'surface

transport'.

P2L20: I feel that either 'age of air' should all be in quotes, or that quotes should not be used. Additionally, throughout the manuscript different the authors use variously age tracer, age of air and age of air tracer. Where possible, it would benefit the reader to use one consistent term.

P2L31: Define TTL

P2L33: Change 10S-10N to include degree symbols to be consistent with elsewhere in the manuscript

P3L10: Please state which reanalysis was used for this study.

P3L15: Here and elsewhere, more care should be taken to stress that it is stratospheric circulation that is being examined.

P3L29: remove 'the' from 'the polar ozone'

P4L2-17: What are the resolutions of the datasets (model and reanalyses) used in the study? What are the impacts of any differences in the resolutions, particularly with regards to mixing?

P4L5: why was only one ensemble member used? What are the expected differences between the ensemble members?

P4L6: Change observe to observed

P4L14: change beneath to below

P4L14-16: I found this sentence confusing and suggest it is reworded.

P4L30: remove an on

P5L8: remove as follows

P5L24: What is meant by steady state here? This term is usually in reference to

chemical change.

P6L4: Is there a need for 'and cooling'? Cooling is just a negative heating.

P6L10: remove naturally

P6L14-16: is there a reference for this statement or is this result calculated for this study?

P7 Figure 1: Is it possible to add contour labels to the correlation figures (also figs 2 and 3) to aid the interpretation of the figures?

P7L2: it would be more accurate to say 'observed tracer distributions' rather than 'tracer measurements'

P7L9: consider changing the use of 'observations' – the authors make the point that one of the problems with the TEM is that it is not observed.

P8L6-7: What is the sensitivity to the choice of latitude bands used here? How does this compare to 10S-10N, the latitude range used earlier in the study for other metrics?

P8L8: what is meant by 'at least 4 times daily data'? 6 hourly data? Are these instantaneous values or means? Similarly for the monthly data – presumably means are required?

P8L22: What is the cause of the difference between the reanalysis and the model for the role of gravity wave drag?

P9L15-16: What is the cause of the changes to r values when using data with different temporal resolutions?

P12L4: Change ERA-I to ERA-Interim to be consistent with the text elsewhere in the manuscript. Also, please check through the manuscript for 'JRA 55', which is sometimes written with a space and sometimes not.

P15L3: Would it be possible to use total hydrogen (H2O+2*CH4) to alleviate the problems encountered due to CH4 oxidation?

P17L10: lower case 't' after ':'

P17L1-15: Throughout the ozone section there is no discussion of ozone chemical lifetime. Many of the results discussing O3 and the branches of the BDC are surely a result of the differences in O3 chemical lifetime at different altitudes? There are recent papers looking at projections of tropical ozone which highlight the role of dynamics in the lower stratosphere and chemistry in the upper stratosphere, and base this distinction on O3 lifetime.

P17L17: Please define what is meant by 'stratospheric entry levels'

P20L7: remove an 'on'

P20L20: Please expand on what is meant by 'can have complications with convergence'. I feel more detail is required on this, either here or in section 4.

---

## Short Comment (SC1) · 18 Dec 2018

Dear authors,

Thank you very much for an interesting study. This comment is motivated by your discussion of the possible influence of the vertical shift of the circulation (isentropes in this case) on the trends of the overturning circulation. (P5aroundL30 and P12L11-12).

I am just submitting a study (Šácha et al., 2018) concerning the effect of the vertical shift of pressure levels (shrinkage) on AoA trends in REFC2 CCMI-1 simulations and so, it was easy for me to have a quick look on the vertical shift of isentropes. Fig. 1 shows a trend (geopotential meters/decade) of isentropic levels in a Ref (1960-2000), NF (2000-2050) and F (2050-2100) period in a CMAM REFC2 simulation. The method is based on interpolation of the standard geopotential height output to predefined isentropic levels using the collocated potential temperature. The isentropes rise quite uniformly at a rate of about 50 meters/decade. However the rise relative to pressure levels will be more rapid higher in the stratosphere, because from about 35km the pressure levels shift downward instead of upward (stratospheric shrinkage). The shift of pressure levels is not shown, because it is included in Šácha et al. (2018) and I am not sure if I can actually show the figure also here.

[Figure]

Figure 1. CMAM trend of geopotential height of isentropic level [gpm/decade]. Only the trends in the regions where they exceed the statistical significance of 95% confidence level are plotted.

Of course, the seasonal trends are different, which further complicates things. For example in DJF (Fig. 2) we have strong significant trends of isentropic levels also in polar regions (interesting for you can be the strong and negative trend a the SH polar region in the Ref period).

[Figure]

Figure 2. The same as Fig. 1 but for DJF.

I am fully aware that your study is based on different dataset and scenario, but I hope that you will find this SC usefull. Please apologize the late submission in the final day of the discussion.

Best regards,

Petr Šácha.

Reference:

Petr Šácha, Roland Eichinger, Hella Garny, Petr Pišoft, Simone Dietmüller, Laura de la Torre, David Plummer, Patrick Jöckel, Olaf Morgenstern, Guang Zeng, Neal Butchart, and Juan Añel: Extratropical Age of Air trends and causative factors in climate projection simulations, ACPD - CCMI special issue.

---

## Author Comment (AC1) · 27 Feb 2019

The comment was uploaded in the form of a supplement:
https://www.atmos-chem-phys-discuss.net/acp-2018-972/acp-2018-972-AC1-supplement.pdf

---

## Author Response (AR1)

This study uses the recently introduced new metric of the BDC, the global overturning diabatic circulation, and investigates its relationship to more traditionally used metrics of calculating the BDC (i.e. different TEM residual circulation metrics). The authors show this relationship for a state of the art climate chemistry model (WACCM) and also for three different reanalysis data sets (ERA-Interim, MERRA, JRA-55). Comparing the different TEM metrics to the global overturning diabatic circulation, they found mainly good agreement in the middle and higher stratosphere, while in the lower stratosphere the difference between the methods is substantial (with highest differences in the reanalysis products). Moreover this paper includes a very nice analysis about the correlation of the diabatic circulation with water vapor tape recorder and also with total column ozone. The results are well organized and described and the topic is appropriate certainly of interest for ACP. I recommend publication with consideration of the specific comments below.

We appreciate your thorough review and interesting questions.

Specific comments

Pg. 2, line 18: '... transport processes such as mixing'. Please include some literature here. Dietmuller et al. 2017, 2018, Ray et al. 2010, 2016.

Pg. 2, line 25: You use different terms for 'global overturning diabatic circulation' in the text, e.g. total overturning circulation, total global diabatic circulation, diabatic overturning circulation, diabatic circulation, global average overturning circulation. Perhaps it is easier for the reader to use the same term in the entire text.

Thanks! Both of the other reviewers have also pointed out this inconsistency, and we now use "global diabatic circulation" throughout.

Pg. 4, line 4: What is the horizontal resolution of the model? We now include a table with a description of the model, reanalysis, and observational products used.

Pg. 4, line 26: I do not understand how the QBO influences the correlations between diabatic circulation and other TEM calculations? As I understand, the QBO should have the same influence on the interannual variability of all BDC metrics, and thus it should not influence the correlations, or? Based on both this comment and the point made by reviewer #2 about the QBO influencing the correlations, it is clear that our treatment of the QBO's influence was insufficient. We have used the word "dominated" instead of "driven" to clarify that we mean correlation not actually physical driving. We have now also included a coherence plot for the correlation of the upper and lower branches (new Figure 1, panel c) as an example of how the QBO is important for the correlation at ~2-year periods, but that there is coherent variability at 8-3 month periods as well. For comparisons between metrics, the coherence is high at all frequencies, and we now state as much:

"A note about the QBO: although the QBO influences both the residual circulation and the global diabatic circulation, the relationships between these metrics are significantly coherent at all frequencies (see Figure 7 for a comparison of timeseries of  $\sum \{w\}^{*}$  and  $\sum \{M\}^{*}$ )" We now address the QBO more explicitly in our discussion of ozone as well, and we hope this has clarified the issue of the way the correlation is impacted by the QBO.

Pg. 5, line 1: Tropical Leaky Pipe Model: Perhaps you could shortly explain the TLP model. Good idea! Done:

"The Tropical Leaky Pipe Model (Neu and Plumb, 1999), a three-box model of the stratospheric circulation that treats the tropics as largely isolated from the extratropics, results in the conclusion that the difference between midlatitude age and tropical age is related to the circulation. Linz et al. (2016) translated this model into isentropic coordinates to show a direct relationship between the idealized age tracer (Hall and Plumb, 1994) and the diabatic circulation through an isentropic surface, demonstrating that the difference between the age of air that is downwelling and the age of air that is upwelling through each isentrope is inversely proportional to the diabatic mass flux through that surface, in steady state and neglecting diabatic diffusion."

Pg. 5, line 6: It would be nice to have an additional sentence about the advantage that the global mean overturning circulation can be assessed from observational data (so you can refer to the statement made at Pg. 4, line 32).

Yes, added:

"Thus, the global diabatic circulation reflects the total tracer flux and should be considered an alternative, or at least an additional, metric. This global diabatic circulation can also be calculated from satellite data." Pg. 7, line 9: Reword to '... primary diagnostic of the stratosphere for models and observations'. And what do you mean with TEM diagnostic from observations?

Thanks for noticing this sloppy language. Now reworded to 'primary diagnostic of the stratosphere for models'

Pg. 8, line 6: Can you explain why you use 30N/S as latitude band? Do you know how sensitive the calculations are if you vary the latitude band to 20N/S or to turn around latitudes?

We used 30N/S as it is relatively common and a more straightforward computation. It is always well defined throughout the depth of the stratosphere, while turnaround latitudes are not (e.g. when there are multiple latitudes of zero crossing). When we checked the WACCM model at 50 hPa, the variability was essentially identical (r=0.95). However, this comment--and the same question was asked by all three reviewers--inspired us to look more closely, and only in the midstratosphere is this relationship so strong. We have added substantially to the discussion, including additional panels and a new figure specifically on this topic.

Pg. 8, line 22: Do you know the reason why it is important for the model and not for reanalysis? No. In the reanalyses, the GWD is very small compared to the model. Investigating the details of why that might be the case is beyond the scope of this paper. We have added to the parenthetical: "(but not in the reanalyses, where the gravity wave drag is much smaller)"

Pg. 8, line 35: Could you mention how the radiative heating was calculated in Rosenlof (1995), e.g. with a radiative transfer model?

"This is consistent with the results of \citet{Rosenlof1995}, who speculated that the reason for this low level discrepancy was the relatively simple way the radiative heating was calculated, using the radiative transfer code developed for two dimensional models by \citet{Yang1991} and \citet{Olaguer1992}."

Pg. 9, line 16: Could you explain, why the correlation is worse when calculated with higher frequency data? What does the study of Ming et al. say about this issue?

This is an interesting result that we haven't dug into in depth. Ming et al. 2016 only examine the impact on the mean and not the variability. If you think of the correlation between the full downward control calculation (with du/dt) and the residual circulation as a statement of how well conservation of momentum applies to the data, then the weaker correlation implies that there are small torques missing from the budget at high frequencies. Other potential reasons would be that the calculation of the tropical velocity (within 180 of the equator) is done by calculating the velocity over the rest of the region and assuming the total residual circulation through any level is 0—at high frequencies, there may be additional storage terms that apply as the pressure surfaces move in the vertical. Since digging into the details of differences for higher frequency calculations is beyond the scope of the paper, we have addressed this as follows:

"We speculate that the worse agreement at higher frequencies is related to either small scale torques that are not captured by the momentum budget at these high frequencies or due to the assumption of instantaneous net-zero flow through each pressure surface, which cannot account for short-term storage." Pg. 12, line 2: It would be easier for the reader, if you could repeat the time period for which the trends are calculated (i.e 1980-2014) here? It was only mentioned in section 2. Added:

"We calculate the trends (1980-2014) in the global diabatic circulation..."

Pg. 12, line 4: Abalos et al. 2015 do not look exactly at the same time period (1979-2012) when looking at trends. Moreover QBO and ENSO variability were removed in Abalos et al. 2015. Is it possible that these facts could also explain the mentioned differences in the trends?

We have now added the parenthetical comment: "(Note Abalos et al. 2015 found that the removal of interannual variability does not change the long-term trends significantly.)"

Pg. 12, line 9: You mention, that isentropic levels are changing their location over these decades. Did you check this for the data you are using?

We looked at it with WACCM, and Petr Šácha, has looked at it in CMAM (comment in the online discussion). There is a continuous trend associated with the changing thermal structure, as one expects with the warming of the troposphere and cooling of the stratosphere.

Pg. 14, figure 6: Why was the correlation only done with w\*, and not with the other TEM residual circulation metrics?

The comparison with the other metrics is pretty much redundant in light of their autocorrelations, and as the w\* is the most common method, we use it alone. However, we have added a third panel that shows the difference when turnaround latitudes are used.

Pg. 15, line 13: Can you explain, why you do not use ozone concentrations from the reanalysis? Correlations of reanalysis data (shown in Fig. 7a-c) would perhaps become better, when the data are more consistent.

Since, as reviewer #2 points out, much of the ozone section is essentially a consistency check, we did not think it necessary to look at the ozone in the reanalyses. The point was simply to demonstrate that the same pattern exists in the model as does with the observations, and so we can use the model to dig in a bit further and make sure the diabatic circulation is behaving consistently with what we might expect.

Pg. 16, line 2: Perhaps you could add a sentence about, how correlation of ozone with the TEM calculations do look like? Or is their a reason why you didn't look at these correlations?

Figure 1 Correlation of total column ozone at each latitude with the tropical upwelling velocity w\* (turn around latitudes) at each level in the WACCM model. Correlations shown are significant at the 95% confidence level.

Again, this comes down to the purpose of the section, which is to satisfy our curiosity of how closely the global diabatic circulation is related to ozone. We always write "The BDC is important for ozone distributions," and it's been shown that the BDC, as measured by w\*, is important, though not as much for variability outside of the QBO (as Reviewer #3 points out). To our knowledge, this hasn't been shown explicitly with the global diabatic circulation strength, and that is the purpose of the section. We've included the correlation plot here, if you are curious. Mostly you see the stronger correlations at the upper level, while the correlations at lower levels are weaker.

Pg. 16, figure 7: I am not sure, but

perhaps an additional plot of the vertical profile of the diabatic circulation and the latitudinal distribution of the O3 column would be nice, to have an idea how they look like. I see what you're saying, but I think focusing on the mean and drawing attention away from the variability would be confusing.

Pg. 17, line 17: Could you define stratospheric entry levels?

Typo! Thanks for catching. It's now changed to "stratospheric entry latitudes".

Pg. 17, line 30: Can you explain why cooling leads to more ozone production? (Or is that clear to everyone?) Changed to say "longer ozone chemical lifetimes", which more accurate. The production, of course, stays the same as it's just a function of actinic flux.

Pg.18, figure 9: Please give the correlation value within the plot (as done in figure 5)? Done.

Pg. 20, line 3: '... metrics for the strength of the BDC. In particular, we have examined ..... and the total column ozone concentrations.' Ozone column is not a metric for strength of BDC. Please reword the sentence.

"In this study, we have compared the global diabatic circulation to the more typically used metrics for the strength of the BDC and to tracers."

Pg. 20, line 19: '...., which can have complication with convergence.'. Can you explain? "Its calculation is simpler than that of  $bar\{w\}^{*}_{Q}$ , which requires some assumption about how to enforce mass conservation  $citep\{Abalos2012\}$ , and which can have complications with convergence when the iterative solving method converges but then occasionally proceeds to diverge after additional iterations."

Pg. 20, line 30: '... in observing systems.' Some more explanation would be nice or give a relevant citation. (Simmons et al. 2014)

Technical corrections

Pg. 2, line 1: Change 'gases' to 'trace gases' Done.

Pg. 2, line 4: 'surface circulation' – Do you mean surface climate?

We meant tropospheric circulation and have fixed accordingly.

Pg. 2, line 11: Perhaps you can reword this sentence, I did have problems to understand it.

Good point. Now rewritten: "Multimodel comparisons (Butchart et al., 2010) and inter-reanalysis comparisons (Abalos et al. 2015, Kobayashi and Iwasaki, 2016) have used the residual mean circulation at 70 hPa, averaged in the tropics, as a metric to evaluate the mean and trends of the BDC."

Pg. 2, line 20: Change to 'age of air' Done.

Pg. 2, line 26: Perhaps change to 'stratospheric circulation strength' Done.

Pg. 2, line 31: 'TTL' – tropical tropopause layer (TTL)

Since we never again refer to the TTL, we have just eliminated the abbreviation entirely and used the words.

- Pg. 2, line 33: '10S-10N' Degree signs are now included (for consistency)
- Pg. 3, line 5: stratospheric circulation Done.
- Pg. 3, line 6: Remove 'it' Added dashes to the appositive for clarity.
- Pg. 3, line 15: stratospheric circulation Done.
- Pg. 4, line 5: Spelling: 'prescribed observed sea surface temperature' Done.
- Pg. 4, line 7: Change to 'model simulation'. Done.
- Pg. 4, line 17: Change to 'heating rates'. Done.
- Pg. 4, line 30: 'circulation on isentropes'. There was one 'on' too much. Done.

Pg. 5, line 1: Perhaps change to 'age of air tracer' Done.

Pg. 6, line 2: Change to '...is outputted differently'. We prefer the English past participle.

Pg. 7, line 19: Change to 'zonal mean wind' Zonal mean zonal wind is correct. The zonal mean of the meridional wind is not used.

- Pg. 8, line 7: Change to '...at all levels.' Done.
- Pg. 9, line 2: 'Abalos et al.:' the year of the citation is missing Added 2015
- Pg. 9, line 6: Replace 'plots' with panels. Whole section has changed
- Pg. 9, line 21: Change to '...is weak'. Done.
- Pg. 12, line 4: 'ERA-Interim' Done.

Pg. 12, line 8: Change to '...for the vertical residual velocity' Changed to "for the global diabatic circulation" It does have a trend in the residual velocity, actually.

Pg. 15, line 3: Change to 'before mentioned' We prefer the use of one word even if it is a bit old-fashioned.

Pg. 17, line 2: Change to '...couple of individual ...' Changed to "two"

Pg. 17, line 24: Perhaps change to Figure 9(a) shows ...' This is optical easier to read. Done.

Pg. 17, line 25: Perhaps include ' ....weaker negative relationship' Done

Pg. 19, line 8: Change to '... with reanalysis and model data' Done

Pg. 20, line 6: Remove one 'on' in '....based on the ...' Done

Anonymous reviewer #2

The goal of this paper is to improve understanding of the global stratospheric diabatic circulation through isentropes (M) as a metric for the Brewer-Dobson Circulation, by making comparisons to other more commonly-used metrics, including derived tropical upwelling and circulations estimated from water vapor and ozone. The calculation and use of M has certain theoretical advantages to diagnose stratospheric transport in an integrated manner, and it is a good idea to make systematic comparisons to other BDC metrics that are commonly used in the research community. These comparisons can pave the wave for more widespread use of M as a diagnostic tool, as proposed in this paper. I especially like the combination of analyzing reanalysis data sets in tandem with results from a self-consistent chemistry-climate model. This paper will make a valuable contribution and the topic is appropriate for ACP. While I strongly endorse the goals of the paper, I have a few major comments on the current content, where I think the paper could be improved prior to publication:

We very much appreciate the thoughtful review. Your comments have highlighted a number of deficiencies in the original manuscript, especially as regards the treatment of the QBO. In the new Figure

1, an example coherence plot is now shown, as are the seasonal cycle of the global diabatic circulation and two deseasonalized timeseries. As the number of figures (and panels!) was already somewhat unwieldy, we chose to highlight select quantities and processes that were particularly enlightening rather than including seasonal cycle and timeseries plots for all of the different diagnostics.

1) The current paper focuses on interannual variability in all of the circulation diagnostics. While this is certainly interesting, I suggest also including comparisons of the actual seasonal cycles in various quantities (climatological monthly time series at a few different theta/pressure levels), which can then serve as a context and background for evaluating interannual variability. In order to enhance the understanding of M, it might be useful to include some simple, approximate conversions of the mass flux to equivalent upwelling velocity, to facilitate direct comparisons to the various estimates of tropical upwelling (w\*, w\*m, w\*Q, wTR). I expect there will be reasonable overall agreement (with, e.g., a large annual cycle in the lower stratosphere).

While an investigation of the seasonal cycle of all of the quantities considered here (global diabatic circulation, three different methods of calculating w\* and now two different latitudes, water vapor tape recorder, and ozone) could be interesting, much of it would be recreating existing figures (e.g. the seasonal cycle of the six versions of w\* for these three reanalyses--18 in total--are Figure 7 of Abalos et al. 2015, seasonal cycles of ERA-Interim w\* and wTR from MLS are compared in Glanville and Birner 2017, Figure 4). Beyond just showing these quantities, however, performing a detailed investigation of differences and similarities in the annual cycles would be a substantial undertaking, beyond the scope of this paper. We agree, however, that the seasonal cycle is useful context for the interannual variability, and so we now show the seasonal cycle of the global diabatic circulation, which has not been shown before. This is in panel a of the new figure 1 (new text about figure 1 is below). In order to see a comparison of the variability of w\* and the global diabatic circulation, w\* is now included with wTR timeseries plot. This should provide an example of the direct comparisons that the reviewer would like.

2) Most of the interannual variability in the results is obviously due to the quasi-biennial oscillation (QBO); this is clearly seen in the time series in Figs. 5 and 9, and the ozone results in Figs. 7-8. In the previous version of the manuscript, these three sentences were buried at the end of the methods section: "In the stratosphere, correlations might be expected to be driven by the Quasi-Biennial Oscillation (QBO) in addition to the seasonal cycle. Rather than explicitly removing this, we account for it by examining filtered time series and cross-spectra (not shown) and highlight the cases where this is important. Many of the relationships examined are coherent with zero phase lag at all frequencies resolved by the monthly time step for the tracers."

We now have made this its own paragraph at the end of the methods section and expanded the discussion. "In the stratosphere, correlations might be expected to be driven by the Quasi-Biennial Oscillation (QBO) in addition to the seasonal cycle. Rather than explicitly removing this, we account for it by examining filtered time series and coherence (e.g. \ref{fig:museas}) and highlight the cases where this is important. Many of the relationships examined are coherent at timescales shorter than the 2-3 year QBO period, though coherence is particularly high at that frequency. The relationship of dynamical variables with trace gases have less high frequency variability, and therefore tend to be dominated by the QBO."

Also, at the beginning of the discussion of the TEM diagnostics, we have included the sentence: "A note about the QBO: although the QBO influences both the residual circulation and the global diabatic circulation, the relationships between the metrics in this section are significantly coherent at all frequencies (see Figure \ref{fig:muh2o} for a comparison of timeseries of  $\lambda = \frac{w}^{*}$  and  $\lambda = \frac{M}{s}$ .)"

We have included a coherence plot for one example that we expected to be entirely QBO driven but found was not: the anticorrelation between the upper and lower branches of the circulation. The timeseries of these are now shown in Figure 1b and coherence in Figure 1c. The correlation is not due entirely to the

QBO—the coherence for periods less than ~9 months is also quite high. We describe the new Figure 1 as follows:

"The seasonal cycle, which is subsequently removed, is shown in the first panel of Figure \ref{fig:museas} for two different levels for the global diabatic circulation from WACCM. The lower stratosphere has a single peak, while the upper stratosphere has a semi-annual cycle as well. This climatology is subtracted to obtain the timeseries shown in the lower panel of Figure \ref{fig:museas}. Note that the negative anomaly is plotted for the lower level, to enable a clear comparison. The different timescales of variability are visible, with an obvious QBO signal and shorter timescale variability. Although the correlation between the upper and lower levels is clear and in phase at QBO timescales, the higher frequency variation is also correlated, but with a 20-90 degree phase lag (not shown). The coherence between these two timeseries is shown in the right panel of Figure \ref{fig:museas}. There is high coherence at periods of 2-3 years, as expected with the QBO. There is also coherence for periods of shorter than about 9 months, which is unexplained by the QBO."

This understanding should be folded into the discussions on comparing the behavior of M and various circulation statistics. For example, the vertical out-of-phase behavior between the lower and upper stratosphere is closely tied to the QBO vertical structure. The patterns of ozone variability (out-of-phase in altitude and latitude) and coupling to meridional circulation are well-known aspects of the QBO signal in ozone (e.g. Bowman, 1989, JAS; Zawodny and McCormick, 1991, GRL; Chipperfield et al, 1994, GRL; Randel et al, 1999, JAS; Tian et al, 2006, JGR).

You are absolutely right that the results for ozone should be put in the context of the known QBO correlations. I have done that as follows:

"Generally, there is a consistent pattern across all three reanalyses and the model. This pattern is consistent with the ozone variability associated with the QBO: an out of phase relationship between the lower and upper stratosphere and an out of phase equatorial and subtropical pattern (e.g. \citet{Zawodny1991})."

"We see that the high latitude total column ozone is correlated with the circulation in the lowermost stratosphere, with the correlation explaining up to 25\% of the deseasonalized total column ozone variability in the Northern hemisphere polar region. The total column ozone in the tropics is strongly anticorrelated with the global diabatic circulation around 500 K. Both of these are qualitatively consistent with transport being the dominant factor driving the relationship between the ozone and the circulation at these levels. The correlation is strongest in the Southern hemisphere in the collar region of the polar vortex, around 55\$^{\circ}\$S, and is weaker at the pole, while in the Northern hemisphere, the correlation is stronger poleward of that, around  $70^{\circ}$  More air is transported by the global diabatic circulation and mixing to the Northern hemisphere pole than the Southern hemisphere pole because the Southern hemisphere polar vortex is a stronger barrier to mixing. The tropical total column ozone is also correlated with the circulation at upper levels, above the ozone maximum (800 K). Like water vapor, the correlation is related to the QBO, and is strongest at ~2 year periods (not shown). Some coherence at higher frequencies can be explained through the anticorrelation of the upper and lower branches of the circulation discussed above. The subtropical total column ozone is anticorrelated with the upper level circulation strength, with hemispheric asymmetry in which levels relate to the subtropical ozone in the different hemispheres. This is consistent with previous results showing the meridional pattern associated with the OBO at these levels leads to opposite anomalies in the deep tropics and the subtropics (e.g. \citealp{Randel1999,Tian2006})."

There are a couple other places now, including the conclusion, where this is brought up briefly.

Also, the in-phase vs. out-of-phase ozone-temperature relationships in the lower and upper stratosphere, respectively, are a well-known general result tied to transport and photochemistry. Yes, this is why it's nice that the diabatic circulation reproduces what we know from our understanding of w\*.

While the M comparisons with the various tropical upwelling estimates were novel and interesting to me, I found the results on ozone (Section 6) to be less valuable for evaluating M as a circulation diagnostic (more of a consistency check with previous results).

It's very much a consistency check. We now state this in the introduction:

"The ozone results are consistent with known relationships between TEM vertical velocity and ozone, demonstrating that the global diabatic circulation is as good a metric for ozone variability."

We felt that for us to endorse adoption of this metric, it should be clear that little change of intuition or understanding is necessary.

Minor comments:

1) In addition to the auto- and cross-correlation diagnostics (Figs. 1-3), it would be valuable to explicitly compare time series of the interannual anomalies in all of the circulation diagnostics, like those included for M and wTR in Fig. 5 (perhaps for time series in the lower and upper stratosphere). This very much helps the reader understand the variability that is quantified in the correlation diagnostics (and provides a 'feel' for the variability among the different diagnostics). Are these comparisons sensitive to the choice of latitude band for the various w quantities?

Again, rather than including all of the circulation diagnostics, we have opted to include a few more timeseries. In the new Figure 1, the timeseries of upper and lower level global diabatic circulation are shown and we have added w\* to the water vapor tape recorder plot in order to show its covariance with the diabatic circulation. As to sensitivity, yes, as mentioned above, the latitude band choice does impact the results, contrary to our preliminary (cursory) examination. An analysis of the turnaround latitudes is now included.

2) P. 5, line 32: you might include a reference to Abalos et al, 2017, JAS, in regards to trend sensitivity to a tropopause-based vertical coordinate.

This is definitely relevant, and now we've actually included a reference to this where we discuss the WACCM trends. See 7 below.

3) It would be good to add a few contour labels to the panels in Figs. 1-3 and 6.

Done.

4) P.8, lines 25-28: the 'downward control' calculations integrate the wave driving multiplied by density, so that in practice the forcing is usually dominated by nearby levels in the vertical (not the entire depth of the stratosphere).

Yes, this is true. The nearby levels (above) impact the circulation at a given level. However, no other method of calculation directly includes information from above or below, so we expect a broader autocorrelation in this case than in those cases. The sentence now reads:

"The downward-control method means that upper levels directly impact lower levels (through integration), and so it is consistent that the vertical autocorrelation of  $\sum \{w\}^{*}_{M}$  is the broadest of all metrics."

5) You might note that w\*Q calculations near the tropopause have an uncertainty in the calculations due to neglect of eddy transport terms (Abalos et al, 2012, ACP). Is this what is meant by 'complications with convergence' (p. 20, line 19)?

We don't mention the eddy transport terms explicitly, but they are now implicitly included in the discussion, and we have clarified what we meant by 'complications with convergence' as follows:

"Its calculation is simpler than that of  $bar\{w\}^{*}Q$ , which requires some assumption about how to enforce mass conservation  $citep\{Abalos2012\}$ , and which can have complications with convergence when the iterative solving method converges but then occasionally proceeds to diverge after additional iterations. Eddy terms in the thermodynamic equation are neglected in the calculation, which may be a reason for these convergence difficulties." 6) The dashed lines relating potential temperature and pressure levels in Figs. 3 and 6 are calculated for an ideal gas, and I guess you mean an isothermal ideal gas. Why not just use a relationship derived from climatological mean values, including realistic temperature structure?

This choice was made because the conversion between pressure and potential temperature is not the same across the globe, and it's not clear whether a global average should be used (because of M) or a tropical average (because of w\*). This ambiguity means that we don't necessarily expect the correlation to fall exactly on whichever relationship we choose. However, exact agreement is implied when a climatological "one-to-one" line is included (and in discussion of these plots, deviations from the climatological theta to p line got more attention than they deserved). To avoid the confusion that the highest correlations should necessarily fall along the line, the relationship for an ideal gas was used as a heuristic.

Now, instead, we have included a statement to explain why deviations from the one-to-one line are not a concern. This should be sufficient to enable use of the climatological values, which are averaged between 20S-20N:

"The climatology of the potential temperature--pressure relationship in the tropics is shown in the dashed line. Note that because the diabatic circulation reflects the global circulation while vertical velocities are calculated only in the tropics, the highest correlations are not necessarily expected to be along this line, but it is a useful visual guide."

The gray lines are now the climatological values for each product.

7) I was surprised to see no significant trends in the WACCM diabatic circulation in Fig. 4, given that many models (including WACCM) show small positive trends in tropical upwelling (e.g. Garcia and Randel, 2008, JAS). What do trends in the various WACCM w\* quantities look like? If these are different from the WACCM M trends, why is that? Is the QBO variability accounted for in these trend calculations?

There are significant trends in the thermal structure, as expected with global warming, that lead to the total circulation through each isentrope remaining constant. Although one might still expect to see an acceleration, it is going to be much weaker ( $\sim$ 4x), as shown by Abalos et al. 2017 with e90 and the residual streamfunction. That paper considered a 145 year WACCM run, while here we only look at 35 years, so it is not surprising that no trend is visible. We have added a sentence:

"This is consistent with the results of \citet {Abalos2017}, who found that trends in the residual streamfunction for a run from 1955-2099 were far weaker when calculated with respect to the tropopause (though the trends were still significantly positive over that long period)."

Removing the QBO does not impact the trend over these multidecadal timescales.

8) I do not understand the overlapping 3-level correlation calculations used to derive wTR from the WACCM water vapor fields. Why is such a complicated calculation necessary? How sensitive are the results to different methods of calculation? How does the background annual cycle of wTR compare with the other upwelling estimates (see major comment 1 above).

After trying a few different methods on a tape recorder where the effective velocity was known, we found a 3-level correlation worked the best at picking up inter-annual variability (like the QBO). The 2-level method is not as good at picking up such variability, but both methods give similar annual mean values. "This method of calculation improves the representation of interannual variability (like the QBO) compared with a simpler two-level method."

9) P. 18, line 4: variations in ozone and (potential) temperature are positively correlated in the lower stratosphere because of similarly signed vertical gradients (and long ozone lifetimes), not because of ozone production.

Yes. You are correct—this is too low for production. The sentence now reads:

"Because it is dynamically controlled, the lower level ozone depends as much on latitude as on the inverse of temperature; the slope is determined by the relative vertical gradients of temperature and ozone."

10) P. 20, line 17 and 19: do you mean w\*Q instead of w\*? Yes.

11) P. 20, lines 19-30: as noted above, the vertical anti-correlation of the interannual circulation diagnosed here is mainly attributable to the QBO vertical structure (linked to tropical wave dynamics and mean flow interactions). This important aspect should be incorporated into the interpretation and summary discussions of vertical structure. Hopefully we have addressed this sufficiently:

"This pattern might be expected with the QBO, but as the coherence is not just at QBO frequencies, an additional mechanism is necessary."

**Anonymous Referee #3**

Received and published: 19 November 2018

Linz et al. present a comparison of the global overturning diabatic circulation with a range of other metrics used to assess stratospheric circulation. They do this using both modelled and reanalysis data. The analysis and discussion presented in the paper is of a high standard and explores an important and relevant topic within the scope of ACP, and as such merits publication following revision.

Thank you for your kind and thoughtful review. The revised manuscript is much clearer because of your comments.

I have several comments the authors should address before publication: General Comments:

1. The authors use a large number of terms to refer to stratospheric circulation in general and the global overturning diabatic circulation in particular. I feel it would aid the reader if consistent terms were used throughout the manuscript. We agree and have done this.

2. Care should be taken when discussing the effects of transport on the distribution of ozone in both sections 1 and 6. It is misleading to say that ozone is produced in the tropics and moved to high latitudes by stratospheric circulation. Brewer and Wilson (1968) and more recently Grewe (2006) highlight that while chemical ozone production in the tropics is high, so is chemical destruction. Grewe (2006) conclude that the view of the tropical region as the global source for stratospheric ozone is highly questionable and that while the tropics tribute to extra-tropical stratospheric ozone, of far greater importance is the production of ozone in the extra-tropics.

The reviewer highlights an important point. The ozone discussion was not explicit about the feedbacks and was ambiguous about what features we are examining (i.e. the variability—not including the seasonal cycle—and not the mean). The use of the word 'primary' was also inaccurate. Hopefully these are addressed by the rewriting of the introductory ozone paragraph (new parts are bold):

"One of the primary motivations for studying the BDC and its variability is its influence on stratospheric ozone. The circulation is known to transport ozone---this is why Dobson proposed it in the first place \citep {Dobson1929}, even if he concluded that this circulation was far-fetched. While the qualitative description of the influence of the stratospheric circulation on ozone **variability** is well established----transport of ozone from its **maximum** production location in the middle stratosphere in the tropics to the midlatitudes and poles---quantifying this effect is not simple. **Furthermore, the interplay between the temperature, ozone and circulation can lead to complex feedbacks.** We know from observational studies that changes to the dynamics impact polar ozone \citep {Hassler2011}, and that the ozone hole recovery is currently being modulated by the dynamics \citep {Solomon2016}. **In turn, variability and trends in the ozone affect the circulation (e.g. \citealp{Polvani2011,Bandoro2014}).** In the Northern hemisphere, the variability in hemispherically averaged upward Eliassen-Palm (EP) flux at 100 hPa from the early NCEP reanalysis data product has been shown to explain about 50\% of the interannual variability of total column ozone in wintertime \citep {Fusco1999} with the influence of the wave driving dependent on the latitude \citep {Reinsel2005}. These strong relationships are a motivating factor in using the TEM residual mean vertical velocity, which is directly related to the EP flux, as a metric for the BDC

strength. The global diabatic overturning circulation on isentropes has been demonstrated to be related to tracer transport more directly, but its relationship with ozone is unknown."

Based on this comment and the comments of Reviewer #2, the ozone results section has changed significantly as well.

3. I miss in the introduction any discussion on the drivers of BDC change or the feedbacks between stratospheric transport and chemical tracers. For example, recent model studies have shown that both GHG increases and polar ozone depletion

accelerate the BDC, while polar ozone recovery may to some extent offset an acceleration of the BDC expected from future GHG increases. Of particular importance to this study, these processes have been shown to affect different branches of the BDC (e.g. Braesicke et al., 2014). Some discussion on how these processes change both the speed and morphology of the BDC may aide in interpretation of the correlations presented in the manuscript.

We do examine trends in the global diabatic circulation in this study. However, the investigation here is not primarily about the morphology of trends, as the global diabatic circulation (as currently defined) cannot distinguish between the two hemispheres. We have now included a brief summary of some of the trend research, as suggested by the reviewer:

"Models predict that the residual mean circulation through a given pressure surface will increase in the future by about 2\% per decade in the lower stratosphere and about 1\% per decade in the middle and upper stratosphere \citep {Butchart2010}. This is a natural consequence of the lifting of the atmospheric circulation (e.g. \citealp {Singh2012,Oberlander2016}), and there are also dynamical reasons why one might expect a true acceleration of the BDC (e.g. \citealp {McLandress2009,Shepherd2011,Garny2011}). However, observations have not shown such a robust trend (e.g.

\citealp{Engel2017,Stiller2012,Haenel2015}). This disagreement can be attributed partially to the large internal variability in the system that prevents a 2\% per decade trend from being detected without 30 years of data \citep{Hardiman2017}, and partially to the fact that there is no truly ``like-to-like" comparison; a modeled tracer that is sampled like the observations can also fail to show a trend even when such a trend exists in the model \citep{Garcia2008}. Models also show that polar ozone loss has dampened the acceleration of the circulation, with an asymmetric effect on the different hemispheres \citep{Polvani2018}."

Additionally, with a focus on the ozone section, changes to the BDC will alter the distribution of radical source gases, in turn altering stratospheric ozone, which will in turn alter the dynamics. We are uncertain what the reviewer is referring to. The radical source gas for ozone is O1D, which is produced by photolysis. The BDC variability is not going to impact the photolysis rates. Perhaps the reviewer is trying to make the point that BDC variability will affect the distributions of N2O, CH4, CFCs, HCFCs, halons, etc..., which when chemically destroyed will impact inorganic NOx, HOx, CIOx, and BrOx species abundances. These abundances will then catalytically affect ozone balance. Understanding this feedback is certainly beyond the scope of this work, and we hope that they are satisfied with our recognition of the complexity.

Highlighting the complexity of the coupled dynamical-chemical system and elaborating on how these feed backs operate would in my view strengthen the introduction and prepare the way for the discussion that follows.

You are right. This interesting coupled system is a major motivation for understanding the BDC. We have now added the sentence in the introduction: "Furthermore, the interplay between the temperature, ozone and circulation can lead to complex feedbacks."

Specific comments:

P1L7: The authors could state here which reanalyses and model is used.

To avoid having to spell out all of the acronyms and add dramatically to the length of the abstract, we prefer to leave this as not specific. We have, however, added a table of the data products used to make this information easier to find at a glance.

P1L14: insert space between 500 and K. Done

P2L4: perhaps change 'surface circulation' to 'tropospheric circulation' or 'surface transport'. Yes, changed to 'tropospheric'

P2L20: I feel that either 'age of air' should all be in quotes, or that quotes should not be used. Additionally, throughout the manuscript different the authors use variously age tracer, age of air and age of air tracer. Where possible, it would benefit the reader to use one consistent term.

Yes, thank you. We've now tried to make these consistent.

P2L31: Define TTL. Done. Good catch.

P2L33: Change 10S-10N to include degree symbols to be consistent with elsewhere in the manuscript Done.

P3L10: Please state which reanalysis was used for this study. It's the early NCEP reanalysis, now specified.

P3L15: Here and elsewhere, more care should be taken to stress that it is stratospheric circulation that is being examined. Here we have added "stratospheric" before "circulation".

P3L29: remove 'the' from 'the polar ozone' Done.

P4L2-17: What are the resolutions of the datasets (model and reanalyses) used in the study? What are the impacts of any differences in the resolutions, particularly with regards to mixing?

We now include a table that shows the resolutions of the model and reanalyses.

We have not explored the impact of resolution on the different metrics. Because it is the diabatic circulation, differences in mixing should not impact the primary diagnostic. They may be more important for the other metrics however. To our knowledge, no systematic comparison of adiabatic mixing with varying resolution has been performed.

P4L5: why was only one ensemble member used? What are the expected differencesbetween the ensemble members? Only one ensemble member was used because the cross correlations and autocorrelations will be robust across members—they are quite robust to removing a few years on either end of the simulation, for example. The only difference one would expect from using a different ensemble member would be in the trend calculation.

P4L6: Change observe to observed. Done.

P4L14: change beneath to below

Thank you for noticing the inconsistency. We prefer 'beneath' and we chose to leave it here and throughout the manuscript. We changed the one instance of 'below' so that we are consistent.

P4L14-16: I found this sentence confusing and suggest it is reworded.

Yes, it was confusing. Now it reads:

Beneath 10 hPa, \citet{Abalos2015} showed that more uncertainty arose from the choice of method of calculation of the vertical velocity than from the choice of reanalysis, concluding that differences between reanalyses were relatively small (except for trends).

P4L30: remove an on. Whoops! Thanks for noticing this.

P5L8: remove as follows

The intent of the sentence was unclear. Its purpose is to introduce the section that provides the mathematical definition of the global diabatic circulation. We have therefore reworded to "The time-dependent, global, diabatic overturning mass flux through an isentrope is defined to be the average of the upwelling and downwelling mass fluxes, as follows.

**As in \citet{Linz2016}, we..."**

P5L24: What is meant by steady state here? This term is usually in reference to chemical change. Steady state means that d/dt=0. We have added "statistically" before the steady state to clarify. P6L4: Is there a need for 'and cooling'? Cooling is just a negative heating.

No need, technically, but it seems clearer.

P6L10: remove naturally. Done

P6L14-16: is there a reference for this statement or is this result calculated for this study? This was a calculation performed for this study.

P7 Figure 1: Is it possible to add contour labels to the correlation figures (also figs 2 and 3) to aid the interpretation of the figures? Absolutely. Done.

P7L2: it would be more accurate to say 'observed tracer distributions' rather than 'tracer measurements' Done.

P7L9: consider changing the use of 'observations' – the authors make the point that one of the problems with the TEM is that it is not observed.

Now reworded to 'primary diagnostic of the stratosphere for models'

P8L6-7: What is the sensitivity to the choice of latitude bands used here? How does this compare to 10S-10N, the latitude range used earlier in the study for other metrics?

There is significant sensitivity to choice of latitude bands between 30N-S vs another reasonable choice, the turnaround latitudes. The 10S-10N latitude is only relevant to the water vapor, which needs to be calculated within the deep tropics to avoid the effects of diffusion at the tropical tropopause. There are now additional panels and paragraphs that address the sensitivity of 30N-S vs. the turnaround latitudes.

P8L8: what is meant by 'at least 4 times daily data'? 6 hourly data? Are these instantaneous values or means? Similarly for the monthly data – presumably means are required?

Changed to '6-hourly', and 'monthly mean'. Then in the next line "For the purposes of this study, the same frequency of data (6-hourly instantaneous values) ..."

P8L22: What is the cause of the difference between the reanalysis and the model for the role of gravity wave drag?

In the reanalyses, the GWD is very small compared to the model. Investigating the details of why that might be the case is beyond the scope of this paper. We have added to the parenthetical: "(but not in the reanalyses, where the gravity wave drag is much smaller)"

P9L15-16: What is the cause of the changes to r values when using data with different temporal resolutions?

So, if you think of the correlation between the full downward control calculation (with du/dt) and the residual circulation as a statement of how well conservation of momentum applies to the data, then the weaker correlation implies that there are small torques missing from the budget at high frequencies. Another potential reason would be that the calculation of the tropical velocity (within 180 of the equator) is done by calculating the velocity over the rest of the region and assuming the total residual circulation through any level is 0—at high frequencies, there may be additional storage terms that apply as the pressure surfaces move in the vertical. Since digging into the details of differences for higher frequency calculations is beyond the scope of the paper, we have addressed this as follows:

"We speculate that the worse agreement at higher frequencies is related to either small scale torques that are not captured by the momentum budget at these high frequencies or due to the assumption of instantaneous net-zero flow through each pressure surface, which cannot account for short-term storage."

P12L4: Change ERA-I to ERA-Interim to be consistent with the text elsewhere in the manuscript. Also, please check through the manuscript for 'JRA 55', which is sometimes written with a space and sometimes not. Thanks for noticing this. They should all be consistent now.

P15L3: Would it be possible to use total hydrogen (H2O+2\*CH4) to alleviate the problems encountered due to CH4 oxidation?

This is an interesting idea and could be a fruitful future research direction. Currently, however, there is potentially concern about using 2\*CH4 across models and data. In a brief examination of H2O created by CH4 oxidation in an old version of WACCM, it was found to be much less than 2\*CH4. In observations, in contrast, there is also an apparent lack of conservation of H2O + 2\*CH4, with values of the CH4 to H2O ratio that are significantly greater than 2 (Wrotny et al. 2010).

P17L10: lower case 't' after ':' now a 'b', but fixed the case

P17L1-15: Throughout the ozone section there is no discussion of ozone chemical lifetime. Many of the results discussing O3 and the branches of the BDC are surely a result of the differences in O3 chemical lifetime at different altitudes? There are recent papers looking at projections of tropical ozone which highlight the role of dynamics in the lower stratosphere and chemistry in the upper stratosphere, and base this distinction on O3 lifetime.

Thank you for bringing this to our attention. The ozone lifetime is implicitly included in the discussion of Figures 9 and 10, which show the effects you refer to. Figure 10 (now Figure 12) shows that at upper levels, the relationship of O3 and temperature "is consistent with a form for many of the reaction rate coefficients for ozone loss processes (Stolarski et al. 2012)."

We now mention it explicitly:

"Ozone variability at upper levels is dominated by photochemical processes \citep {Perliski1989}, resulting in a short chemical lifetime, and so we hypothesize that this close correlation is due to the relationship of temperature with both ozone and the circulation strength. When the circulation is stronger in the tropics at these levels, that is associated with cooling and consequently longer ozone chemical lifetimes."

P17L17: Please define what is meant by 'stratospheric entry levels'

Yes, good catch. We meant 'stratospheric entry latitudes'.

P20L7: remove an 'on' Done.

P20L20: Please expand on what is meant by 'can have complications with convergence'. I feel more detail is required on this, either here or in section 4.

Thanks for noting this. "Its calculation is simpler than that of  $bar\{w\}^{*}Q$ , which requires some assumption about how to enforce mass conservation  $citep\{Abalos2012\}$ , and which can have complications with convergence where the iterative solving method converges but then occasionally proceeds to diverge after additional iterations."

**The global overturning diabatic circulation of the stratosphere as a metric for the Brewer-Dobson Circulation**

Marianna Linz1, Marta Abalos2, Anne Sasha Glanville3, Douglas E. Kinnison3, Alison Ming4, and Jessica L. Neu5 1Department 
[revised manuscript text omitted]

**3.2 Global diabatic circulation characteristics**